# Repairing Neural Networks by Leaving the Right Past Behind

**Ryutaro Tanno**[*1]     **Melanie F. Pradier**[1]     **Aditya Nori**[1]     **Yingzhen Li**[2]
[1]Microsoft Health Futures, Cambridge, UK     [2]Imperial College London, UK

## Abstract

Prediction failures of machine learning models often arise from deficiencies in training data, such as incorrect labels, outliers, and selection biases. However, such data points that are responsible for a given failure mode are generally not known a priori, let alone a mechanism for repairing the failure. This work draws on the Bayesian view of continual learning, and develops a generic framework for both, identifying training examples which have given rise to the target failure, and fixing the model through erasing information about them. This framework naturally allows leveraging recent advances in continual learning to this new problem of model repairment, while subsuming the existing works on influence functions and data deletion as specific instances. Experimentally, the proposed approach outperforms the baselines for both identification of detrimental training data and fixing model failures in a generalisable manner.

## 1   Introduction

Machine learning (ML) models often exhibit unexpected failures once deployed in the "wild". Recent lines of research aim to alleviate different well-known shortcomings in supervised models, such as vulnerability to annotation errors [1] and adversarial attacks [2], sensitivity to data shifts [3], and biases to underrepresented subgroups [4, 5, 6]. However, it is often challenging to anticipate beforehand all plausible failure scenarios, and protect against them pre-emptively. This motivates developing a technique that is able to *repair a model on demand*, as new failure cases arise in practice.

Undesirable behaviours of ML models commonly stem from defects in the training data. However, it is unclear how to detect the causes of such failures automatically, rendering a manual troubleshooting necessary. Furthermore, once the problems are uncovered, one would still need to design fixes, which typically involve further data curation/collection, and model retraining/redesigning from scratch. Executing the above steps demand not only time, but also mature expertise in the relevant ML areas, a scarcity in the present job market.

This work introduces an approach to identifying a set of most detrimental training examples that have caused failure cases observed at test time, and to subsequently repairing the model on these failures by deleting those culprits. At the basis of both cause identification and repairment steps is the approximation of *"counterfactual" posterior distribution* where some training examples are assumed absent. We formalise this as a Bayesian *continual (un)learning* problem [7], where the above counterfactual posterior is estimated by deleting the evidence of selected training data from the current posterior. We note that, while other factors (e.g. model class and optimisation) may play a role, we focus on "data debugging" and investigate, to what extent, prediction failures could be remedied by only intervening on data and updating the model accordingly.

Fig. 1 gives an overview of the proposed approach for model repairment, which operates in two steps by 1) identifying causes of failure among training data, and 2) updating the model by erasing the "memories" of those harmful examples. Importantly, the proposed framework is agnostic to *why* a particular datapoint hurts model performance, handling both, issues in the input data and/or labels. We only require specification of the set of failed cases for which we wish to improve performance.

---

[*]Now at DeepMind, UK. Correspondence to `rtanno@google.com` and `yingzhen.li@imperial.ac.uk`

36th Conference on Neural Information Processing Systems (NeurIPS 2022).

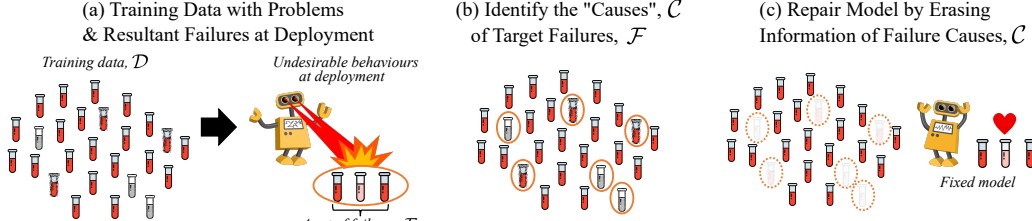

(a) Training Data with Problems & Resultant Failures at Deployment

(b) Identify the "Causes", $\mathcal{C}$ of Target Failures, $\mathcal{F}$

(c) Repair Model by Erasing Information of Failure Causes, $\mathcal{C}$

Figure 1: (a) Real-world datasets are often fraught with issues such as annotation noise, low-quality inputs, anomalies, and acquisition biases (e.g. demographic imbalances). Such issues may lead to undesirable performance of the trained models in deployment. Our approach aims at repairing such models by (b) identifying detrimental training examples which have caused the target failures, and then (c) erasing efficiently the *memories* of those examples from the models.

**Our contributions:** We develop a framework for *repairing* machine learning models by erasing memories of detrimental datapoints. The framework connects both identification and removal of detrimental data under a (Bayesian) continual learning perspective, which brings forth practical benefits. Firstly, the framework subsumes works on influence function [8] and data deletion [9] as specific examples, which are developed independently, and our work reveals their close connections and limitations. Secondly, the generality of our formulation allows translating any continual learning method into this model repairment setting, and opens doors to further research. In particular, we extend Elastic Weight Consolidation [10] – a specific continual learning algorithm – to cause identification and data removal, and demonstrate improvements over the prior works in a variety of settings where training data are contaminated with annotation and/or input noise.

## 2 Model Repairment by Data Deletion

Let us consider a prediction model $p(y|\boldsymbol{x},\boldsymbol{\theta})$ that returns a probability distribution of the output $y$ given an input $\boldsymbol{x}$. We make the i.i.d. modelling assumption and denote $p(\boldsymbol{\theta}|\mathcal{D})$ as the posterior distribution over the model parameters $\boldsymbol{\theta}$ given training data $\mathcal{D}=\{(\boldsymbol{x}^{(n)},y^{(n)})\}_{n=1}^{N}$. At test time, the posterior predictive distribution is used to infer label $y^{\star}$ given a new sample $\boldsymbol{x}^{\star}$:

$$p(y^{\star}|\boldsymbol{x}^{\star},\mathcal{D})=\int p(y^{\star}|\boldsymbol{x}^{\star},\boldsymbol{\theta})p(\boldsymbol{\theta}|\mathcal{D})d\boldsymbol{\theta}, \tag{1}$$

where $p(y^{\star}|\boldsymbol{x}^{\star},\boldsymbol{\theta})$ is the likelihood term for sample $(x^{\star},y^{\star})$. While we use approximate posteriors in practice, we focus on the exact case for now to formalise the problem.

Imagine that, in deployment, this model makes incorrect predictions in certain situations. After collecting a "failure set" $\mathcal{F}=\{(\boldsymbol{x}_f^{(n)},y_f^{(n)})\}_{n=1}^{N_f}$ of examples from such failure mode in the test set[2], we would like to *repair* the model, such that it improves performance on the failure set $\mathcal{F}$ and similar future cases. We also argue that a successful repairment should also *maintain* a similar level of performance on the rest of test examples. These two objectives of model repairment are analogous to those of a *medical treatment*, in that both aim to fix a specific problem while leaving the "healthy" part as intact as possible. A further discussion of different aspects, including generalisation, efficiency and specificity properties is provided in Appendix A.1.

In this work, we assume that the main reason for such failures $\mathcal{F}$ is due to the existence of detrimental examples in the training data $\mathcal{D}$, for example, noisy labels, low-quality inputs and/or group imbalances. Our hypothesis is that, by removing these harmful datapoints and adapting the model accordingly, the model can be repaired to return correct predictions for datapoints in $\mathcal{F}$ as well as similar future cases. We acknowledge that there might be other reasons for a model to make wrong predictions on $\mathcal{F}$, such as bad local optima and model biases, which are outside the scope of this work. Addressing data-based failures is complementary to accounting for other problems and is of relevance regardless, as datasets typically come with unexpected issues no matter how much we curate them beforehand.

We formulate the process of **model repairment** as the following two steps (see Fig. 1 for an illustration):

1. **Cause identification:** Identify a set of detrimental datapoints, i.e. "failure causes" $\mathcal{C}$ in the training data $\mathcal{D}$ that contributed the most to the failure set $\mathcal{F}$.

2. **Treatment:** Given the set of failure causes $\mathcal{C}$, adapt the model to predict correctly on the failure set $\mathcal{F}$, while maintaining performance on remaining test examples.

---

[2]There may be multiple different ways in which the model fails [11], and the failure cases may consist of several groups. One type of mistake might incur more costs than others (e.g. in certain medical applications, *false negative* is more costly than *false positive*). Here we assume that we have identified at least one failure type that we would like to fix.

---

**Algorithm 1** Model Repairment

---

**Input:** training data $\mathcal{D}$; failure cases $\mathcal{F}$; approximate posterior $q(\boldsymbol{\theta}) \approx p(\boldsymbol{\theta}|\mathcal{D})$; likelihood $p(\boldsymbol{z}|\boldsymbol{\theta})$
**Output:** failure causes $\mathcal{C}$, "repaired" posterior $q_{-\mathcal{C}}(\boldsymbol{\theta})$
# Step I: Cause Identification
**Update posterior:** Apply a *continual learning* method to obtain $q_{+\mathcal{F}}(\boldsymbol{\theta}) \approx p(\boldsymbol{\theta}|\mathcal{D}, \mathcal{F})$ by fitting the failure set $\mathcal{F}$
**Compute influences of training examples on $\mathcal{F}$:** Calculate $\tilde{r}(\boldsymbol{z}) \forall \boldsymbol{z} \in \mathcal{D}$ (Eq. (9))
**Find failure causes $\mathcal{C}$:** Return the examples with positive influence, $\mathcal{C} \leftarrow \{\boldsymbol{z} \in \mathcal{D}: \tilde{r}(\boldsymbol{z}) > 0\}$
# Step II: Treatment
**Delete information of $\mathcal{C}$:** Apply a *continual (un)learning* method to the original posterior $q(\boldsymbol{\theta})$, and obtain the posterior on the corrected data $q_{-\mathcal{C}}(\boldsymbol{\theta}) \approx p(\boldsymbol{\theta}|\mathcal{D} \backslash \mathcal{C})$

---

Below we describe a formal framework for performing these two steps. In the first phase of cause identification, we need to define a measure of how much a subset of training examples $\mathcal{C} = \{(\boldsymbol{x}_c^{(n)}, y_c^{(n)})\}_{n=1}^{N_c} \subset \mathcal{D}$ is responsible for the failure cases $\mathcal{F}$. To this end, we propose to check how much the posterior predictive distribution on $\mathcal{F}$ changes as a result of deleting $\mathcal{C}$ from the training data:

$$r(\mathcal{C}) := \log p(\mathcal{F}|\mathcal{D} \backslash \mathcal{C}) - \log p(\mathcal{F}|\mathcal{D}), \tag{2}$$

where $p(\mathcal{F}|\mathcal{D})$ and $p(\mathcal{F}|\mathcal{D} \backslash \mathcal{C})$ are the posterior predictive distributions before and after removing a subset of training examples $\mathcal{C}$ defined as follows:

$$p(\mathcal{F}|\mathcal{D}) = \int p(\mathcal{F}|\boldsymbol{\theta}) p(\boldsymbol{\theta}|\mathcal{D}) d\boldsymbol{\theta}, \; p(\mathcal{F}|\mathcal{D} \backslash \mathcal{C}) = \int p(\mathcal{F}|\boldsymbol{\theta}) p(\boldsymbol{\theta}|\mathcal{D} \backslash \mathcal{C}) d\boldsymbol{\theta}, \; p(\mathcal{F}|\boldsymbol{\theta}) = \prod_{(\boldsymbol{x},y) \in \mathcal{F}} p(y|\boldsymbol{x}, \boldsymbol{\theta}). \tag{3}$$

If a given $\mathcal{C}$ leads to a large positive value of $r(\mathcal{C})$ in Eq. (2), it means that removing $\mathcal{C}$ from $\mathcal{D}$ would have improved the performance of the Bayesian predictive inference on the failure set $\mathcal{F}$. The first *cause identification* step thus entails finding a subset $\mathcal{C}$ with the maximal log-density ratio $r(\mathcal{C})$. In the second *treatment* step, we can directly adopt $p(\mathcal{F}|\mathcal{D} \backslash \mathcal{C})$ for such identified $\mathcal{C}$ as the updated predictive distribution, as it confers the largest improvement over the failure cases. In other words, the posterior predictive distribution $p(\mathcal{F}|\mathcal{D} \backslash \mathcal{C})$ after data removal is key to both the search of detrimental datapoints and repairment of the model.

The central computational question is, therefore, concerned with the *efficient calculation of $p(\mathcal{F}|\mathcal{D} \backslash \mathcal{C})$ without retraining the model from scratch.* The reasons for avoiding retraining are: (1) in the cause identification step, it is computationally prohibitive as retraining needs to be done for every subset $\mathcal{C} \subset \mathcal{D}$; and (2) in the treatment step, the resulting retrained model may be drastically different from the original model one would like to fix (due to, e.g. noises in the SGD-based optimisation and parameter re-initialisation), and may lose other desirable properties that one may wish to retain.

In this work, we present a continual learning [12] framework to simultaneously address the aforementioned computational challenges in cause identification and treatment. The framework is summarised in Algorithm 1, with the following key developments by leveraging (Bayesian) continual learning:

1. For **cause identification** (Section 2.1), we present a fast approximation to $r(\mathcal{C})$, which requires a one-off approximation of $p(\boldsymbol{\theta}|\mathcal{D}, \mathcal{F})$ only via continual learning, and enables a *linear-time* search for the detrimental datapoints in $\mathcal{C}$.

2. For **treatment** (Section 2.2), we show that the approximation of $p(\boldsymbol{\theta}|\mathcal{D} \backslash \mathcal{C})$ for $\mathcal{C}$ identified in the first step can be achieved by performing a new "continual learning task" using $\log p(\mathcal{C}|\boldsymbol{\theta})$ as its "loss function".

Our framework is generic and flexible in the sense that any continual learning approach can be applied to both steps; we demonstrate, in particular, a concrete instantiation by leveraging Elastic Weight Consolidation [10] as the base continual learning approach. We now elaborate on the mathematical details of the two steps.

## 2.1 Step I: Cause Identification

Identifying the set of detrimental examples $\mathcal{C}$ requires solving the following optimisation problem

$$\mathcal{C} = \operatorname{argmax}_{\mathcal{C}' \in \mathbb{P}(\mathcal{D})} r(\mathcal{C}'), \tag{4}$$

where $\mathbb{P}(\mathcal{D})$ denotes the power set of $\mathcal{D}$. Solving this comes with multiple computational challenges.

Firstly, a naive approach would require computing the predictive distribution $p(\mathcal{F}|\mathcal{D}\setminus\mathcal{C})$ — and thus, the posterior $p(\boldsymbol{\theta}|\mathcal{D}\setminus\mathcal{C})$ — for every subset $\mathcal{C}$ of $\mathcal{D}$, which is prohibitively expensive. To address this, we present a **"predictive" approach** that removes this computational burden. The key idea is to notice that

$$p(\mathcal{D}\setminus\mathcal{C}|\boldsymbol{\theta})=p(\mathcal{D}|\boldsymbol{\theta})/p(\mathcal{C}|\boldsymbol{\theta}), \quad \forall \mathcal{C}\subset\mathcal{D},$$

due to the i.i.d. modelling assumption. Inserting this into the Bayes' rule for computing $p(\boldsymbol{\theta}|\mathcal{D}\setminus\mathcal{C})$ yields:

$$\log p(\mathcal{F}|\mathcal{D}\setminus\mathcal{C})=\log\int p(\mathcal{F}|\boldsymbol{\theta})p(\boldsymbol{\theta}|\mathcal{D}\setminus\mathcal{C})d\boldsymbol{\theta}=\log\int\frac{p(\mathcal{F}|\boldsymbol{\theta})}{p(\mathcal{C}|\boldsymbol{\theta})}\frac{p(\mathcal{D}|\boldsymbol{\theta})p(\boldsymbol{\theta})}{p(\mathcal{D})}d\boldsymbol{\theta}-\log\frac{p(\mathcal{D}\setminus\mathcal{C})}{p(\mathcal{D})}.$$

Notice that again due to the i.i.d. modelling assumption,

$$\frac{p(\mathcal{D}\setminus\mathcal{C})}{p(\mathcal{D})}=\int\frac{1}{p(\mathcal{C}|\boldsymbol{\theta})}\frac{p(\mathcal{D}|\boldsymbol{\theta})p(\boldsymbol{\theta})}{p(\mathcal{D})}d\boldsymbol{\theta}.$$

Therefore, we can compute the log density ratio $r(\mathcal{C})=\log p(\mathcal{F}|\mathcal{D}\setminus\mathcal{C})-\log p(\mathcal{F}|\mathcal{D})$ in the following "predictive" form, without computing $p(\boldsymbol{\theta}|\mathcal{D}\setminus\mathcal{C})$—see Appendix A.2 for derivations:

$$r(\mathcal{C})=\log\int\frac{p(\boldsymbol{\theta}|\mathcal{D},\mathcal{F})}{p(\mathcal{C}|\boldsymbol{\theta})}d\boldsymbol{\theta}-\log\int\frac{p(\boldsymbol{\theta}|\mathcal{D})}{p(\mathcal{C}|\boldsymbol{\theta})}d\boldsymbol{\theta}=\log\mathbb{E}_{p(\boldsymbol{\theta}|\mathcal{D},\mathcal{F})}[p(\mathcal{C}|\boldsymbol{\theta})^{-1}]-\log\mathbb{E}_{p(\boldsymbol{\theta}|\mathcal{D})}[p(\mathcal{C}|\boldsymbol{\theta})^{-1}]. \quad (5)$$

In this form, we only need to compute the posterior $p(\boldsymbol{\theta}|\mathcal{D},\mathcal{F})$ once, and can side-step the requirement of computing $p(\boldsymbol{\theta}|\mathcal{D}\setminus\mathcal{C})$ for inspection of every new $\mathcal{C}\subset\mathcal{D}$, thereby removing one of the key computational bottlenecks. Intuitively, as the "predictive" formulation (Eq. (5)) of $r(\mathcal{C})$ computes the log expectation difference of $p(\mathcal{C}|\boldsymbol{\theta})^{-1}$, higher $r(\mathcal{C})$ means that datapoints in $\mathcal{C}$ are less likely to be predicted correctly when the posterior is updated by the information from the failure set $\mathcal{F}$. In other words, this indicates conflicting information exists between $\mathcal{F}$ and $\mathcal{C}$, and as we would like to repair the model to produce correct predictions on $\mathcal{F}$ and similar examples, the information of $\mathcal{C}$ should be removed from the model.

Secondly, we are still left with the combinatorial search for the best subset $\mathcal{C}$, which is also prohibitive when the size of training data $\mathcal{D}$ is large, even when the "predictive" formulation Eq. (5) is used. We address this issue by a first-order Taylor series approximation of $r(\mathcal{C})$. Let us re-write the log density ratio as

$$r(\mathcal{C})=F(1,p(\boldsymbol{\theta}|\mathcal{D},\mathcal{F}))-F(1,p(\boldsymbol{\theta}|\mathcal{D})), \quad \text{where} \quad F(\epsilon,g(\boldsymbol{\theta})):=\log\int g(\boldsymbol{\theta})e^{-\epsilon\log p(\mathcal{C}|\boldsymbol{\theta})}d\boldsymbol{\theta}. \quad (6)$$

Note that $F(0,g(\boldsymbol{\theta}))=0$ for any well-defined distribution $g(\boldsymbol{\theta})$. We now perform a Taylor expansion of $F(\epsilon,g(\boldsymbol{\theta}))$ around $\epsilon=0$: $F(\epsilon,g(\boldsymbol{\theta}))=-\epsilon\mathbb{E}_{g(\boldsymbol{\theta})}[\log p(\mathcal{C}|\boldsymbol{\theta})]+\mathcal{O}(\epsilon^2)$. Finally we obtain an approximate log-density ratio $\hat{r}(\mathcal{C})\approx r(\mathcal{C})$ by plugging the first term of the Taylor expansion into the RHS of Eq. (6):

$$\hat{r}(\mathcal{C}):=\mathbb{E}_{p(\boldsymbol{\theta}|\mathcal{D})}[\log p(\mathcal{C}|\boldsymbol{\theta})]-\mathbb{E}_{p(\boldsymbol{\theta}|\mathcal{D},\mathcal{F})}[\log p(\mathcal{C}|\boldsymbol{\theta})]. \quad (7)$$

Assuming that data are i.i.d., and defining $\boldsymbol{z}=(\boldsymbol{x},y)$, the above approximation can be expressed as the sum of individual log density ratios, $\hat{r}(\mathcal{C})=\sum_{\boldsymbol{z}\in\mathcal{C}}\hat{r}(\boldsymbol{z})$, where each term is given by

$$\hat{r}(\boldsymbol{z})=\mathbb{E}_{p(\boldsymbol{\theta}|\mathcal{D})}[\log p(\boldsymbol{z}|\boldsymbol{\theta})]-\mathbb{E}_{p(\boldsymbol{\theta}|\mathcal{D},\mathcal{F})}[\log p(\boldsymbol{z}|\boldsymbol{\theta})], \quad p(\boldsymbol{z}|\boldsymbol{\theta})=p(y|\boldsymbol{x},\boldsymbol{\theta}). \quad (8)$$

Critically, with this approximation, in order to find a subset $\mathcal{C}$ of cardinality $K$ that leads to the maximal $\hat{r}(\mathcal{C})$, it suffices to compute $\hat{r}(\boldsymbol{z})$ for every training example $\boldsymbol{z}\in\mathcal{D}$ and find the top $K$ examples with largest $\hat{r}(\boldsymbol{z})$ values, thereby reducing the search space from $\mathcal{O}(|\mathcal{D}|!)$ choices to only $\mathcal{O}(|\mathcal{D}|)$ choices.

The per-instance formulation Eq. (8) has similar interpretation as the "predictive" formula Eq. (5), in that $\hat{r}(\boldsymbol{z})$ measures how much the predictive moments of $\boldsymbol{z}$ changes when the model is further trained on $\mathcal{F}$, i.e., computation of $p(\boldsymbol{\theta}|\mathcal{D},\mathcal{F})$. If the difference is positive, i.e., $\hat{r}(\boldsymbol{z})>0$, it means the example $\boldsymbol{z}\in\mathcal{D}$ is a conflicting evidence against the test examples in $\mathcal{F}$; conversely, if $\hat{r}(\boldsymbol{z})<0$, then $\boldsymbol{z}$ and $\mathcal{F}$ are aligned. In practice, the failure causes correspond to examples $\boldsymbol{z}$ with $\hat{r}(\boldsymbol{z})>0$.

Lastly, for non-linear models (e.g. neural networks), approximate posteriors $q(\boldsymbol{\theta})\approx p(\boldsymbol{\theta}|\mathcal{D})$ and $q_{+\mathcal{F}}(\boldsymbol{\theta})\approx p(\boldsymbol{\theta}|\mathcal{D},\mathcal{F})$ are needed due to intractability of the exact posteriors. We assume that $q(\boldsymbol{\theta})$ is available after training, and suffers from the prediction failures $\mathcal{F}$. As recomputing the posterior $q_{+\mathcal{F}}(\boldsymbol{\theta})$ from scratch can be expensive, we propose to use a *continual learning* technique [12] and obtain this quantity by updating the original posterior $q(\boldsymbol{\theta})$. Finally, we use the following metric $\tilde{r}(\boldsymbol{z})$ in practice to calculate the detrimental impact of each training datapoint on the failure set, $\mathcal{F}$, by replacing the exact posteriors in Eq. (8) with their corresponding approximations:

$$\tilde{r}(\boldsymbol{z}):=\mathbb{E}_{q(\boldsymbol{\theta})}[\log p(\boldsymbol{z}|\boldsymbol{\theta})]-\mathbb{E}_{q_{+\mathcal{F}}(\boldsymbol{\theta})}[\log p(\boldsymbol{z}|\boldsymbol{\theta})], \quad (9)$$

and the top $K$ entries according to $\tilde{r}(z)$ are selected to approximate the failure causes $\mathcal{C}$. This metric is generic, and its implementation depends on the specifics in which both $q(\boldsymbol{\theta})$ and $q_{+\mathcal{F}}(\boldsymbol{\theta})$ are computed, e.g., MLE/MAP point estimates, Laplace approximation [13], variational inference [14, 15], etc. Two concrete examples are provided: the first shows that the well-known *linear influence function* [8] is a specific instance of Eq. (9); and the second is derived by extending a continual learning method, known as *Elastic Weight Consolidation* (EWC) [10] to cause identification, and is a key methodological development in our work.

***Example 1 (Linear Influence Function):*** Our proposed metric in Eq. (9) recovers the *linear influence function* from Koh & Liang [8] when point estimates are used for $\boldsymbol{\theta}$. Assume that the model is trained on data $\mathcal{D}$ with parameters $\hat{\boldsymbol{\theta}}$, which corresponds to an approximation of MLE/MAP estimates, i.e., $q(\boldsymbol{\theta}) = \delta(\boldsymbol{\theta} - \hat{\boldsymbol{\theta}}) \approx p(\boldsymbol{\theta}|\mathcal{D})$. After observing the set of failures $\mathcal{F}$, a point estimate of $p(\boldsymbol{\theta}|\mathcal{D},\mathcal{F})$ is obtained by performing a single update of natural gradient ascent [16] on the log likelihood of $\mathcal{F}$ with step size $\gamma > 0$:

$$q_{+\mathcal{F}}(\boldsymbol{\theta}) = \delta(\boldsymbol{\theta} - \hat{\boldsymbol{\theta}}_{+\mathcal{F}}) \approx p(\boldsymbol{\theta}|\mathcal{D},\mathcal{F}), \quad \hat{\boldsymbol{\theta}}_{+\mathcal{F}} \approx \hat{\boldsymbol{\theta}} + \gamma \hat{\boldsymbol{F}}_{\hat{\boldsymbol{\theta}}}^{-1} \nabla_{\hat{\boldsymbol{\theta}}} \log p(\mathcal{F}|\hat{\boldsymbol{\theta}}), \tag{10}$$

where $\hat{\boldsymbol{F}}_{\hat{\boldsymbol{\theta}}}$ is the empirical Fisher information matrix. This means

$$\tilde{r}(z) = -\gamma \nabla_{\hat{\boldsymbol{\theta}}} \log p(\mathcal{F}|\hat{\boldsymbol{\theta}})^{\top} \hat{\boldsymbol{F}}_{\hat{\boldsymbol{\theta}}}^{-1} \nabla_{\hat{\boldsymbol{\theta}}} \log p(z|\hat{\boldsymbol{\theta}}), \tag{11}$$

if defining the updated posterior as $q_{+\mathcal{F}}(\boldsymbol{\theta}) = \delta(\boldsymbol{\theta} - \hat{\boldsymbol{\theta}}_{+\mathcal{F}})$. The negation of the above equation coincides with the linear influence function (Eq. (2) in [8]) when the failure set is assumed to be a singleton $\mathcal{F} = \{z_{test}\}$ and $\gamma = 1$. Note that the sign difference arises since our work aims to quantify the "negative" influence rather than the "positive" one in contrast with the work of Koh and Liang [8].

***Example 2 (Elastic Weight Consolidation):*** The generality of Eq. (9) permits any continual learning method of one's choice for estimating the updated posterior $q_{+\mathcal{F}}(\boldsymbol{\theta}) \approx p(\boldsymbol{\theta}|\mathcal{D},\mathcal{F})$ after observing failure samples $\mathcal{F}$. Here we illustrate how EWC [10] as a continual learning method can be adopted in the context of cause identification. EWC approximates $p(\boldsymbol{\theta}|\mathcal{D},\mathcal{F})$ by first performing Laplace approximation of the original posterior $p(\boldsymbol{\theta}|\mathcal{D})$ around the point estimate $\hat{\boldsymbol{\theta}}$, and subsequently finding the MAP solution of $\boldsymbol{\theta}$. Formally, $\hat{\boldsymbol{\theta}}_{+\mathcal{F}}$ is obtained by maximising the objective below w.r.t. $\boldsymbol{\theta}$ via SGD (see Appendix A.3 for details):

$$\log p(\mathcal{F}|\boldsymbol{\theta}) - \frac{N}{2}(\boldsymbol{\theta} - \hat{\boldsymbol{\theta}})^{\top} \hat{\boldsymbol{F}}_{\hat{\boldsymbol{\theta}}}(\boldsymbol{\theta} - \hat{\boldsymbol{\theta}}) - \frac{\lambda}{2}\|\boldsymbol{\theta} - \hat{\boldsymbol{\theta}}\|_2^2, \tag{12}$$

where the off-diagonal elements of $\hat{\boldsymbol{F}}_{\hat{\boldsymbol{\theta}}}$ are dropped for memory reason in practice. Intuitively, the first term encourages high accuracy on the failure set $\mathcal{F}$ whilst the second/third terms ensures the model parameters do not deviate too much in distribution. Defining $q(\boldsymbol{\theta}) = \delta(\boldsymbol{\theta} - \hat{\boldsymbol{\theta}})$ and $q_{+\mathcal{F}}(\boldsymbol{\theta}) = \delta(\boldsymbol{\theta} - \hat{\boldsymbol{\theta}}_{+\mathcal{F}})$, we have that

$$\tilde{r}(z) = \log p(z|\hat{\boldsymbol{\theta}}) - \log p(z|\hat{\boldsymbol{\theta}}_{+\mathcal{F}}). \tag{13}$$

To compute the above for each datapoint $z \in \mathcal{D}$, we only need to solve the optimisation problem of Eq. (12) by SGD once. We refer to this version of $\tilde{r}(z)$ as *EWC-influence function*.

**Comparison:** The EWC-influence function generalises the linear influence approach. To see this, we derive the fixed point of the EWC objective Eq. (12) w.r.t. $\boldsymbol{\theta}$ (where we set $\lambda = 0$):

$$\boldsymbol{\theta} = \hat{\boldsymbol{\theta}} + N^{-1} \hat{\boldsymbol{F}}_{\hat{\boldsymbol{\theta}}}^{-1} \nabla_{\boldsymbol{\theta}} \log p(\mathcal{F}|\boldsymbol{\theta}). \tag{14}$$

Then the update in Eq. (10) that is implicitly used by linear influence function can be viewed as (damped) one-step fixed-point iteration update initialised at $\hat{\boldsymbol{\theta}}$ for solving the fixed-point equation. As EWC-influence update (Eq. (13)) is obtained by using the optimum of Eq. (12), it is arguably more accurate than linear influence function (Eq. (11)) for measuring the (detrimental) effect of a datum $z$ to the model failures $\mathcal{F}$.

## 2.2 Step II: Treatment

Once the causes $\mathcal{C}$ of the failures $\mathcal{F}$ are identified among the training data $\mathcal{D}$, we seek to repair the model $q(\boldsymbol{\theta})$ by erasing the memories of $\mathcal{C}$. We formalise this problem as the computation of the posterior $p(\boldsymbol{\theta}|\mathcal{D}\setminus\mathcal{C})$, i.e., $\mathcal{C}$ is absent from the training data $\mathcal{D}$. A naive approach would re-run approximate inference on the whole "corrected" dataset $\mathcal{D}\setminus\mathcal{C}$ to obtain an approximate posterior $q_{-\mathcal{C}}(\boldsymbol{\theta}) \approx p(\boldsymbol{\theta}|\mathcal{D}\setminus\mathcal{C})$ which can be time consuming. But more fundamentally, by doing so, the obtained $q_{-\mathcal{C}}(\boldsymbol{\theta})$ may be unrelated to the original $q(\boldsymbol{\theta})$ based on which $\mathcal{C}$ were identified, due to, e.g. non-convex SGD optimisation issues. Moreover, this "model replacement" approach may not maintain other good properties of the original model $q(\boldsymbol{\theta})$.

Analogous to cause identification (Sec. 2.1), we propose to employ the continual learning approach to estimate efficiently the modified posterior $p(\boldsymbol{\theta}|\mathcal{D}\backslash\mathcal{C})$. Applying Bayes' rule and some algebraic manipulations yield $p(\boldsymbol{\theta}|\mathcal{D}\backslash\mathcal{C})\propto p(\boldsymbol{\theta}|\mathcal{D})/p(\mathcal{C}|\boldsymbol{\theta})$ (see Appendix A.4). Therefore the information about $\mathcal{C}$ can be removed by scaling the current posterior $p(\boldsymbol{\theta}|\mathcal{D})$ by the inverse of $p(\mathcal{C}|\boldsymbol{\theta})$ and re-normalising. In other words, we can treat the approximation of $p(\boldsymbol{\theta}|\mathcal{D}\backslash\mathcal{C})$ as a continual learning task, where the task is to "unlearn" the datapoints in $\mathcal{C}$ while using the posterior distribution $p(\boldsymbol{\theta}|\mathcal{D})$ as the prior. In practice, the target model to be fixed corresponds to the approximate posterior $q(\boldsymbol{\theta})\approx p(\boldsymbol{\theta}|\mathcal{D})$. Therefore continual (un)learning is done by

$$q_{-\mathcal{C}}(\boldsymbol{\theta})\propto q(\boldsymbol{\theta})/p(\mathcal{C}|\boldsymbol{\theta})\approx p(\boldsymbol{\theta}|\mathcal{D}\backslash\mathcal{C}). \tag{15}$$

The above approximation can be carried out with different approximate inference techniques such as MLE/MAP point estimate, Laplace approximation [10] and variational inference [7]. Again we provide a few examples to concretise this process as follows.

***Example 1 (Fine-tuning on Corrected Data):*** Given a point estimate of model parameters $\hat{\boldsymbol{\theta}}$, i.e., $q(\boldsymbol{\theta})=\delta(\boldsymbol{\theta}-\hat{\boldsymbol{\theta}})$, a simple way to approximate $p(\boldsymbol{\theta}|\mathcal{D}\backslash\mathcal{C})$ is to fine-tune on the corrected dataset $\mathcal{D}\backslash\mathcal{C}$ and update the point estimate. The new $\hat{\boldsymbol{\theta}}_{-\mathcal{C}}$ of the repaired model are obtained by maximising the log-likelihood $\log p(\mathcal{D}\backslash\mathcal{C}|\boldsymbol{\theta})$ via SGD, starting from $\hat{\boldsymbol{\theta}}$.

***Example 2 (Newton Update Removal):*** Guo *et al.*[9] proposed a Newton update based method for data deletion. This method reduces to a specific form of Eq. (15) when using log-likelihood as its loss:

$$\hat{\boldsymbol{\theta}}_{-\mathcal{C}}\approx\hat{\boldsymbol{\theta}}-\gamma\hat{\boldsymbol{F}}_{\hat{\boldsymbol{\theta}}}^{-1}\nabla_{\hat{\boldsymbol{\theta}}}\log p(\mathcal{C}|\hat{\boldsymbol{\theta}}). \tag{16}$$

Here information about $\mathcal{C}$ gets deleted by performing a single-step natural gradient descent on their log likelihood [16]. Also, notice the similarity with the way linear influence [8] is computed in Eq. (10), illuminating the relation between cause identification and treatment steps.

***Example 3 (EWC for data deletion):*** The update rule in Eq. (15) for data deletion is amenable to any continual learning approaches. For example, given model parameters $\hat{\boldsymbol{\theta}}$, EWC-based deletion obtains new parameters $\hat{\boldsymbol{\theta}}_{-\mathcal{C}}$ by maximising the following objective (see Appendix A.4):

$$-\log p(\mathcal{C}|\boldsymbol{\theta})-\frac{N}{2}(\boldsymbol{\theta}-\hat{\boldsymbol{\theta}})^{\top}\hat{\boldsymbol{F}}_{\hat{\boldsymbol{\theta}}}(\boldsymbol{\theta}-\hat{\boldsymbol{\theta}})-\frac{\lambda}{2}||\boldsymbol{\theta}-\hat{\boldsymbol{\theta}}||_2^2 \tag{17}$$

where the first term seeks to remove information about $\mathcal{C}$ while the remaining terms discourage parameters from deviating from the original values. Contrasting this with Eq. (12) again reveals the connection between EWC methods for cause identification and treatment steps.

**Comparison:** Similar to the comparison made in the cause identification part, EWC for data deletion also generalises the Newton update removal (Eq. (16)). This can again be shown by deriving (damped) one-step fixed point iterative update starting from $\hat{\boldsymbol{\theta}}$ to approximate the fixed point of Eq. (17) when $\lambda=0$. As EWC for deletion uses SGD to approximate optimum of Eq. (17), it is arguably better than Newton update removal for erasing the effects of detrimental examples $\mathcal{C}$, while better maintaining performance on other cases.

**Connecting the two steps:** We highlight the deep connection between the "predictive approach" for cause identification (Sec. 2.1), and the continual (un)learning for data deletion in the treatment step. They share the key idea of editing the approximate posterior $q(\boldsymbol{\theta})\approx p(\boldsymbol{\theta}|\mathcal{D})$ via continual learning, which corresponds to editing the factor graph [17] of $p(\boldsymbol{\theta}|\mathcal{D})$, e.g., insertion of $p(\mathcal{F}|\boldsymbol{\theta})$ in cause identification, and deletion of $p(\mathcal{C}|\boldsymbol{\theta})$ in treatment. This unified view enables ones to take any continual learning method and extend it to both steps of model repairment. Indeed in our experiments, EWC as a better continual learning approach than e.g., one-step Newton update leads to improvement in both tasks over the prior works.

## 3 Related work

**Model Editing**. There is a recent surge of interest in developing targeted updates to correct model's undesirable behaviours, while leaving other desired properties intact. As naive fine-tuning methods often lead to overfitting to the failure examples and accuracy degradation on others, various strategies have been proposed. For example, Zhu *et al.*[18] employ a simple regularization technique to minimize parameter changes during the fine-tuning phase. Subsequent works [19, 20] advocate for a functional regularisation instead, e.g. KL divergence in the output space, to achieve better regularisation. These lines of work, additionally, propose to use meta-learning [21] to learn to edit the target model, where the latest meta-learning approach is proposed

by Mitchell *et al.*[22]. Another promising approach [23] performs weight editing so that features of a specific concept (e.g. snow) map to the features of another (e.g. road). A commonality among these approaches is the focus on direct model edits for correction. Our work takes an orthogonal and under-explored angle where the aim is to "edit" the data instead, by identifying and removing harmful examples which cause failures — in turn, this difference makes our framework complementary to these model-editing approaches.

**Continual learning.** Continual learning is an active research area with a related but broader scope than model repairment, which aims to develop methods that adapt the model for future tasks while maintaining model performance on previously learned tasks [12]. We focus on a more targeted problem in this work, yet introduce a framework that allows the use of any continual learning approach for model repairment. Our experiments presents EWC [10] as a practical instantiation of the framework. One can also leverage improvements over EWC such as online EWC [24], or other regularisation-based methods that are motivated by Bayesian learning principles, such as variational continual learning [7, 25, 26], synaptic intelligence [27], and orthogonal gradient descent [28]. As approximations to $r(\mathcal{C})$ rely on accurate posterior approximations, advances in Bayesian continual learning methods are expected to improve the practical effectiveness of model repairment under our framework.

**Data Selection and Valuation**. Multiple techniques have been introduced for selecting "influential" training examples on a chosen metric (e.g. test accuracy), such as influence functions [8, 29, 30, 31, 32, 33, 34], Shapley value-based approaches [35, 36, 37] and probability of sufficiency [38]. Within the category of influence functions, two representative approaches include linear influence function [8] and SGD-influence [32]. The former approach performs one-step update only, thus, while efficient, it may be less accurate in reflecting the influence of a datum $z$. The latter approach computes a projected difference between $\hat{\theta}$ and $\hat{\theta}_{+\mathcal{F}}$ but with $\hat{\theta}_{+\mathcal{F}}$ obtained by running SGD fine-tuning on training data without $z$. Thus SGD-influence is computationally inefficient. Compared to both baselines, our EWC-influence approach achieve the best in both worlds: it produces more accurate influence estimates than linear influence due to better optimisation, while it is more efficient than SGD-influence as it requires only one optimisation procedure on the given failure set $\mathcal{F}$.

**Data Deletion**. The detrimental data removal in the treatment step is related to *data deletion*, a rapidly developing field of machine learning research [39, 9, 40, 41, 42, 43]. Closest to our work is variational Bayesian unlearning [44] which extends variational Bayes to data deletion settings. But the connection to continual learning is not explicitly made, and it is limited to applications in logistic regression and sparse Gaussian processes. In general, the main focus of existing data deletion research is to preserve data privacy, and datapoints to be removed are assumed provided. On the contrary, in this work, we focus on the repairment of models and propose a unified procedure not only to remove data but also to identify which ones to do so.

# 4 Experiments

We evaluate the efficacy of the proposed framework in a) identifying the causes of target prediction failures in Sec. 4.1, and b) repairing the original model by erasing the memories of such causes in Sec. 4.2. We use augmented versions of MNIST and CIFAR-10 datasets with simulated annotation and input noise. Such controlled experiments are performed for creating "ground truths" of failure causes – necessary for validating the quality of identification methods – and for testing the method in a variety of settings.

**Baselines.** For the cause identification task, we compare our approach (*EWC-influence*) against the linear influence function [8] and *SGD-influence* [32]. To avoid expensive computation of $\hat{F}_{\hat{\theta}}^{-1}$, Koh & Liang [8] introduced two efficient approximations to the Hessian-vector product $\hat{F}_{\hat{\theta}}^{-1} \nabla_{\theta} \log p(\mathcal{F}|\theta)$; the first solves $\arg\min_v \{v^T \hat{F}_{\hat{\theta}}^{-1} \nabla_{\theta} v - \log p(\mathcal{F}|\theta)^T v\}$ with gradient descent (GD), while the second uses an iterative algorithm for stochastic approximation (SA) from [45]. We implement these two variants (GD & SA) of linear influence in Pytorch, and use the original implementation for SGD-influence. For the model treatment task, we compare our method (*EWC-deletion*) against Newton update removal [9]. This method again requires computing a Hessian-vector product for which we employ the same stochastic approximation technique [45]. To isolate the evaluation of cause identification and treatment, we further consider in Section 4.1 *fine-tuning* on $\mathcal{D} \backslash \mathcal{C}$ as another repairment strategy, which would return the best repairment result if the set $\mathcal{C}$ correctly captures the detrimental datapoints. Lastly, we set the prior term $\lambda$ to zero in eq.(12) and eq. (17) to ensure fair comparison with linear influence and Newton update removal.

**Common Set-up.** We train the base classification models on the training split of the "augmented" MNIST and CIFAR-10 datasets. For MNIST, we use $6\%$ (3000 samples) of the original training set to make the task

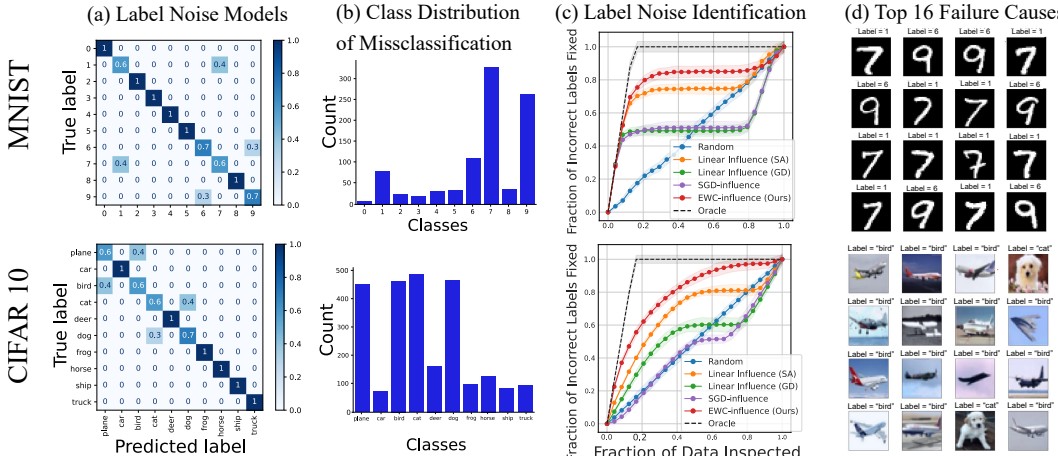

Figure 2: Results on cause identification in the presence of annotation noise. (a) shows the confusion matrices used to simulate class-dependent label noise on MNIST and CIFAR-10. (b) shows the class distribution of the misclassified examples for a single run. (c) plots how much of the identified causes match the samples with incorrect labels for different approaches. The shade represents the standard deviation computed from 5 different runs. (d) shows the top 16 causes of the failures as ranked by EWC-influence. An enlarged version can be found in Appendix D.

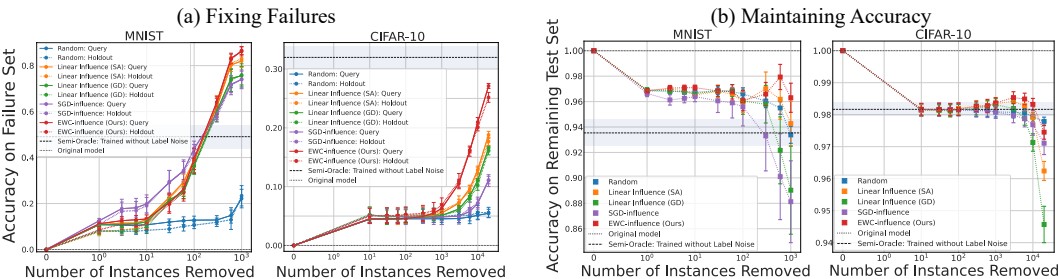

Figure 3: Comparison of the quality of identified causes in the presence of annotation noise. The impact of gradually removing samples in MNIST and CIFAR-10 datasets in the order of influence values $r(z)$ are measured on the failure sets (holdout and query) in (a), and on the remaining test set in (b). We note that in (b), accuracy values start at 1.0 as they are calculated on the set of test samples on which the original model makes correct predictions. We also plot the performance of another reference ("semi-oracle") that is the original model fine-tuned on the training data without the label noise instances. The means/stds of all quantities are calculated over 5 runs. See Appendix D for an enlarged version.

more challenging. We use instances of CNNs throughout and train them using the Adam optimiser [46]. The architecture and training details can be found in Appendix C. For evaluation, we separate the test set $\mathcal{T}$ into the set of misclassified examples, $\mathcal{F}$ ("*failure set*") and the others, $\mathcal{T} \setminus \mathcal{F}$ which are correctly classified ("*remaining set*"). We further split the failure set into *query*, $\mathcal{F}_q$ and *holdout*, $\mathcal{F}_h$ sets, where we only use the former to identify failure causes $\mathcal{C}$, and use the latter to quantify how generalisably the removal of $\mathcal{C}$ can amend the failure cases. We stress that $\mathcal{F}_q$ is used for cause identification only, but not for further model adaptation.

## 4.1 Identifying Failure Causes

**Annotation Noise.** To induce test prediction failures, we randomly flip labels in the training set between semantically similar classes (e.g. 1 and 7 for MNIST, and cats and dogs for CIFAR-10) according to the confusion matrices in Fig. 2(a). As a result, the classes of miss-classified test examples are concentrated on those classes with label noise as depicted in Fig. 2(b).

To measure the accuracy of identifying incorrectly labelled examples, we inspect the training examples $z \in \mathcal{D}$ in the descending order of $\tilde{r}(z)$ computed with $\mathcal{F}_q$ which contains 50% of the miss-classified test cases, and calculate the fraction of incorrectly labelled datapoints in inspected examples. Fig. 2(c) shows that EWC-influence identifies more failure causes earlier on compare to other methods, and is the closest match to the "Oracle" baseline which has full knowledge of samples with wrong labels. Fig. 2(d) shows that the top few causes according to EWC-influence are the samples with incorrect labels, while the least harmful ones are the images of the same classes but with the correct labels as shown in Fig. 7 in Appendix B.

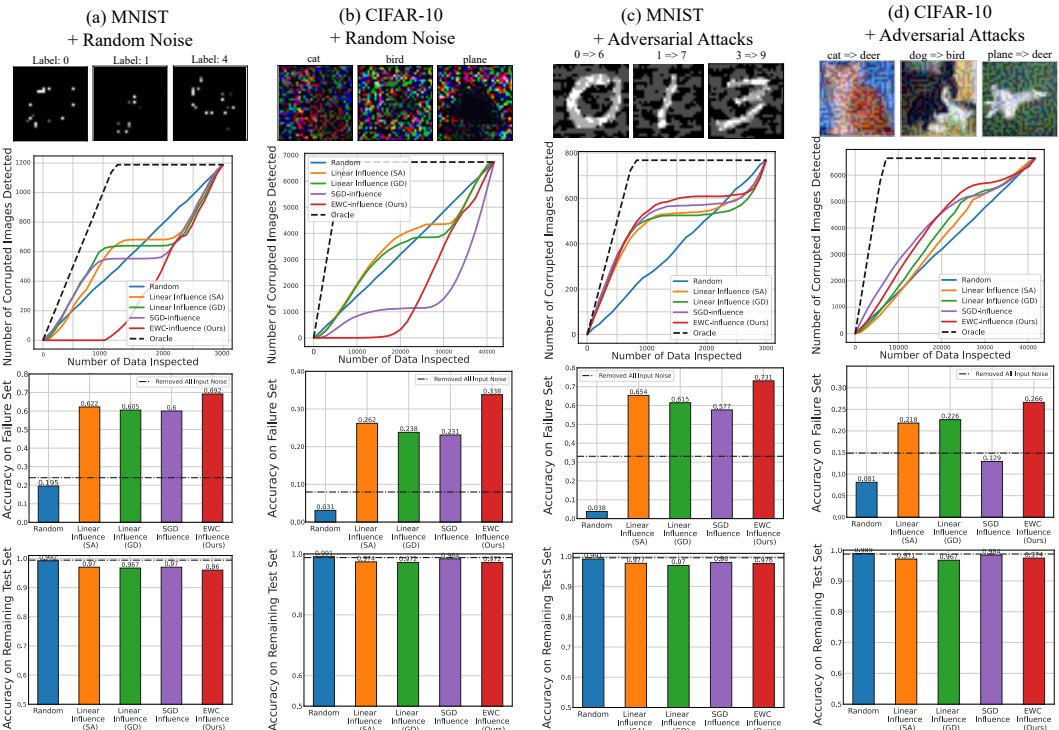

Figure 4: Results on cause identification in the presence of different input noises. From top to bottom, we show i) examples of corrupted samples (synthetic proxy for potential causes of failure), ii) how many of the identified causes correspond to samples corrupted with input noise, iii) and iv) performance in failure holdout set $\mathcal{F}_h$ and remaining test set when removing the top 1000/20000 identified causes in MNIST/CIFAR-10. The influence values are calculated with respect to 50% of test-time failure cases that belong to the classes that suffer from input noise. EWC-Influence identifies "harmful" (adversarial) input noise better than random while avoiding "harmless" (random) input noise. An enlarged version is available in Appendix D.

As stipulated in Sec. 2, a set of identified causes $\mathcal{C}$ is of higher quality if removing them leads to a larger gain in accuracy on the failure set while maintaining performance on others. To measure such *quality of causes*, we fine-tune the base model on $\mathcal{D}\backslash\mathcal{C}$ and report the accuracy on the failure query set $\mathcal{F}_q$, the holdout failure set $\mathcal{F}_h$ as well as the remaining test set, $\mathcal{T}\backslash\mathcal{F}$. Results in Fig. 3 suggest that removing failure causes according to EWC-influence yields the highest increase in accuracy on the failure set $\mathcal{F}$ without hurting performance in the remaining test set $\mathcal{T}\backslash\mathcal{F}$. We also note that all of the methods are able to fix the failures better than randomly removing datapoints, and more interestingly, for MNIST, when enough causes are erased ($\approx 10^3$), all methods even surpass the case in which all label noise instances are removed. This result implies that, while annotation noise is a major detrimental factor, the prediction failures also arise from other types of harmful examples.

Lastly, we evaluate the sample efficiency of cause identification by reducing the size of the query set $\mathcal{F}_q$. Fig. 6 in Appendix B shows that all approaches degrade gracefully in repairment performance as the query size gets smaller, but overall EWC-influence still remains the best in terms of label noise detection and repairment accuracy on the failure set and the remaining set.

**Random Input Noise.** In this experiment, we inject synthetic outliers into MNIST and CIFAR-10 and test the quality of cause identification. We select a set of target classes — 1, 7, 6, 9 for MNIST and plane, bird, cat, dog for CIFAR-10 — and randomly corrupt 30% of the images in those classes by adding salt-and-pepper noise (i.e. replacing pixels with extreme values 0 and 255) in MNIST and Gaussian noise in CIFAR-10. The top row in Fig. 4 shows examples, and those corrupted images constitute roughly 12% of the whole training set. Sec. C in the appendix provides details.

We use a subgroup of failures in the target classes as the query $\mathcal{F}_q$ to compute influence values. Surprisingly, the second rows in Fig. 4(a) and (b) show that EWC-influence largely avoids selecting the corrupted images as the top 1000 causes for MNIST and the top 20000 causes for CIFAR-10. However, the third and the fourth rows show that removing those causes results in the best treatment performance on failures while maintaining the performance at a level similar to other baselines. In fact, removing all the input noise and

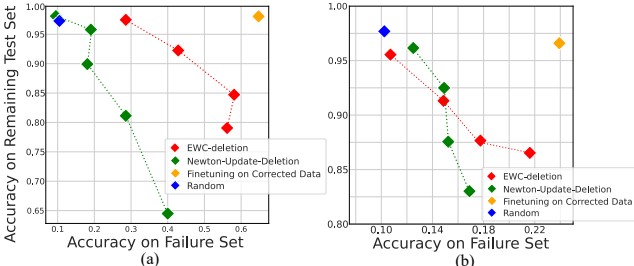

Figure 5: Comparison of deletion-based treatment methods on (a) MNIST and (b) CIFAR-10. For Newton-update-deletion and EWC-deletion, we plot multiple results for varying hyper-parameters to visualise the trade-off between the accuracy on the failure set and the remaining set. The closer to the top right corner, the more desirable.

retraining is not able to fix the failures by much, indicating that EWC-influence is able to correctly avoid these relatively harmless outliers and detect other more harmful causes. Fig. 8 in Appendix B visualises the most harmful examples identified by EWC-influence. Many of them appear to be ambiguous instances in non-target classes, e.g. wonky digits, close-up views of vehicles, a real instance with incorrect label [47], etc.

**Adversarial Poisoning.** To simulate input noise that can induce test-time failures, we introduce contaminated data by randomly corrupting 30% of the training images in those previously mentioned target classes. These poisoned datapoints are adversarial images crafted by the fast gradient sign method (FGSM) [48] on a separate set of victim models trained on the original clean datasets, and they are labelled by the classes predicted by the victim models. The poisoned datasets are then used to train the base models that are used for evaluation of cause identification. Fig. 4(c) and (d) show that most of the influence functions detect the corrupted samples better than the "random" baseline. The dashed lines in the third row show that removing all of the corrupted inputs lead to a significant gain in accuracy on the holdout failure set in comparison with the random noise setting, illustrating the larger extent of harms caused by data poisoning. However, most of the identification methods still outperform this reference by a large margin. This suggests again for the presence of other influential samples, and EWC-influence is able to pick up the most important ones, judging by the accuracy on the failure set.

**Speed Comparison.** Table 1 in Appendix B shows the total run-time of cause identification methods on a single GPU for their best sets of hyper-parameters selected based on the treatment accuracy on the failure set. For both datasets, EWC-influence achieves comparable or shorter run time than the baselines.

## 4.2 Comparison of Treatment Methods

We evaluate the performance of different deletion-based methods for treatment introduced in Sec. 2.2 on MNIST and CIFAR10 datasets with simulated annotation noise, used in the previous section. We run both EWC-deletion (ours) and Newton update removal [9] methods with early stopping based on the query set accuracy, and experiment with different hyper-parameter settings (see Sec. C in Appendix) to achieve different trade-offs between failure set accuracy and remaining set performance. Here EWC-influence is used to identify the causes, and the top 15% examples were removed by the respective deletion methods. Such trade-off is shown in Fig. 5, where fine-tuning on $\mathcal{D}\backslash\mathcal{C}$ is included as an "upper-bound" reference for data deletion performance. On MNIST, EWC-deletion attains a considerably better trade-off between treatment and maintenance compared to Newton-update-deletion, and is much closer to the fine-tuning reference. For CIFAR-10, EWC-deletion beats the Newton-update deletion by 5% in the best failure accuracy while the order reverses for the best accuracy on the remaining test set but with less than 1% difference.

## 5 Conclusions

In this work, we develop a generic framework for *repairing* machine learning models by erasing memories of detrimental datapoints. The framework consists of two key components, that are, the mechanism for identifying the "causes" in training data which are responsible for the given failures, and the adaptation method for fixing the model by removing information about them. The two components are connected under the Bayesian view of continual (un)learning, which brings forth several practical benefits. Firstly, the framework subsumes some recent works on influence function and data deletion as specific examples, and elucidate their limitations. Secondly, the generality of our approach allows leveraging recent advances in continual learning in this new problem of model repairment. In particular, we extend Elastic Weight Consolidation to cause identification and data deletion, and demonstrate empirically its competitive performance in both tasks.

## Acknowledgements

We would like to thank Ozan Oktay, Stephanie Hyland, and Ted Meeds at Microsoft Research Cambridge and Jin Chen at University College London for their valuable feedback on an early version of this work.

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
