# A  Algorithmic details

## A.1  Problem formulation for model repairment

This section extends the discussion on the problem formulation of model repairment. Specifically, under the modelling assumptions presented in the main text, the goals for model repairment are the following:

- For a set of "failure cases" $\mathcal{F} = \{z_f = (x, y)\}$ where the model with (Bayesian) predictive inference makes wrong predictions, repair the model to make correct predictions on $\mathcal{F}$ and similar cases.
- For a set of "benchmark cases" $\mathcal{B} = \{z_b = (x, y)\}$, maintain a given level of prediction accuracy after model repairment.

There are further considerations for executing and evaluating model repairment in practice.

**Generalisation and efficiency of model repairment**  In practice the number of failure cases might be large or even infinite. For example, an image classification model that fails on a test input with Gaussian noise may also fail on all the other inputs with such level of noise. So we need to consider both, *generalisation* and *efficiency* aspects of model repairment. Here, good generalisation means that the failure is fixed, not only for observed failure cases but also for future cases for which the model would make the same type of failure before fixes. On the other hand, efficiency of repairment considers the number of failure examples required for the model repairment method to fix a particular type of failure.

To describe both concepts in more details, in addition to a set of known/observed failure examples $\mathcal{F}$ collected by the users, we need to define the set of *unknown/unobserved* failure examples $\mathcal{F}_U$. If examples in $\mathcal{F}$ and $\mathcal{F}_U$ are similar, then good *generalisation* of a model repairment algorithm means that a model repaired by such method using information from $\mathcal{F}$ should produce correct predictions for instances in $\mathcal{F}_U$. On the other hand, a model repairment method is *efficient* if it only needs a small set $\mathcal{F}$ of collected failure cases to achieve good generalisation of repairment.

**Specificity of model repairment**  Furthermore, there may be multiple different scenarios in which the model fails, and therefore the set of failure cases may consist of several groups, i.e.,

$$\mathcal{F} = \sqcup_{m=1}^{M} \mathcal{F}^{(m)}, \quad \mathcal{F}_U = \sqcup_{m=1}^{M} \mathcal{F}_U^{(m)}, \tag{18}$$

where $\mathcal{F}^{(m)}$ denotes the observed failure cases of failure type $m$ and $\mathcal{F}_U^{(m)}$ represents unobserved failure cases of the same type. Targeting a specific type of mistake and repairing it one at a time may be desirable in practice. One type of mistake might incur more costs than others (e.g., in medical applications, false negative is generally more costly than false positive), so users might have different priorities for different types of errors to be fixed. This is especially the case when there exists a trade-off between fixing different types of errors, again the false negative vs false positive trade-off is a prevalent example.

**Repairment by identifying and removing detrimental training data**  There can be many different reasons for a model with Bayesian predictive inference making wrong predictions on $\mathcal{F}$. In this work, we assume that *the main reason is due to the existence of detrimental datapoints in $\mathcal{D}$*. Our hypothesis is that, by removing/correcting these detrimental datapoints and adapting the model with them, the model can be repaired to return correct labels for datapoints in $\mathcal{F}$. Base on the above hypothesis, the model repairment process contains the following steps:

1. **Cause identification:** Identify a set of detrimental data points $\mathcal{C}$ in the training data $\mathcal{D}$ that contributed the most to the failure set $\mathcal{F}$.
2. **Treatment:** Given the set of failure causes $\mathcal{C}$, adapt the model to predict correctly on the failure set $\mathcal{F}$, while maintaining performance on other examples which were correctly predicted previously.

## A.2  The "predictive approach" for cause identification

We use the following function to describe the contribution of a subset $\mathcal{C} \subset \mathcal{D}$ to the model failures on examples in $\mathcal{F}$:

$$r(\mathcal{C}) = \log\left(\frac{p(\mathcal{F}|\mathcal{D}\backslash\mathcal{C})}{p(\mathcal{F}|\mathcal{D})}\right). \tag{19}$$

A naive approach would compute $p(\boldsymbol{\theta}|\mathcal{D}\setminus\mathcal{C})$ for all subsets of $\mathcal{D}$ with all possible correction methods, which is prohibitively expensive. Instead we present a **"predictive" approach** that removes this computational burden. First, notice that we make the i.i.d. modelling assumption which means that one can write the likelihood term as follows:

$$p(\mathcal{D}|\boldsymbol{\theta}) = p(\mathcal{D}\setminus\mathcal{C}|\boldsymbol{\theta})p(\mathcal{C}|\boldsymbol{\theta}), \quad \forall \mathcal{C}\subset\mathcal{D}. \tag{20}$$

This allows us to expand the log evidence $\log p(\mathcal{F}|\mathcal{D}\setminus\mathcal{C})$ as

$$
\begin{aligned}
\log p(\mathcal{F}|\mathcal{D}\setminus\mathcal{C}) &= \log \int p(\mathcal{F}|\boldsymbol{\theta})p(\boldsymbol{\theta}|\mathcal{D}\setminus\mathcal{C})d\boldsymbol{\theta} \\
&= \log \int p(\mathcal{F}|\boldsymbol{\theta})\frac{p(\mathcal{D}\setminus\mathcal{C}|\boldsymbol{\theta})p(\boldsymbol{\theta})}{p(\mathcal{D}\setminus\mathcal{C})}d\boldsymbol{\theta} \qquad \text{(Bayes' rule)} \\
&= \log \int p(\mathcal{F}|\boldsymbol{\theta})\frac{p(\mathcal{D}|\boldsymbol{\theta})p(\boldsymbol{\theta})}{p(\mathcal{C}|\boldsymbol{\theta})p(\mathcal{D}\setminus\mathcal{C})}d\boldsymbol{\theta} \qquad \text{(by Eq. (20))} \\
&= \log \int \frac{p(\mathcal{D})}{p(\mathcal{D}\setminus\mathcal{C})}\cdot\frac{p(\mathcal{F}|\boldsymbol{\theta})}{p(\mathcal{C}|\boldsymbol{\theta})}\cdot\frac{p(\mathcal{D},\boldsymbol{\theta})}{p(\mathcal{D})}d\boldsymbol{\theta} \qquad \left(\text{multiplying } \frac{p(\mathcal{D})}{p(\mathcal{D})} \text{ and rearranging terms}\right) \\
&= \log \int \frac{p(\mathcal{F}|\boldsymbol{\theta})}{p(\mathcal{C}|\boldsymbol{\theta})}p(\boldsymbol{\theta}|\mathcal{D})d\boldsymbol{\theta} + \log\frac{p(\mathcal{D})}{p(\mathcal{D}\setminus\mathcal{C})}. \qquad \text{(Bayes' rule)}
\end{aligned}
$$

Then we can rewrite the log density ratio as

$$
\begin{aligned}
r(\mathcal{C}) &= \log\left(\frac{p(\mathcal{F}|\mathcal{D}\setminus\mathcal{C})}{p(\mathcal{F}|\mathcal{D})}\right) \\
&= \log \int \frac{p(\mathcal{F}|\boldsymbol{\theta})}{p(\mathcal{C}|\boldsymbol{\theta})}p(\boldsymbol{\theta}|\mathcal{D})d\boldsymbol{\theta} - \log p(\mathcal{F}|\mathcal{D}) + \log p(\mathcal{D}) - \log \int p(\mathcal{D}\setminus\mathcal{C}|\boldsymbol{\theta})p(\boldsymbol{\theta})d\boldsymbol{\theta} \\
&\qquad\qquad \text{(by definition of marginal distributions)} \\[2mm]
&= \log \int \frac{1}{p(\mathcal{C}|\boldsymbol{\theta})}\frac{p(\mathcal{F}|\boldsymbol{\theta})p(\boldsymbol{\theta}|\mathcal{D})}{p(\mathcal{F}|\mathcal{D})}d\boldsymbol{\theta} - \log \int \frac{p(\mathcal{D}|\boldsymbol{\theta})}{p(\mathcal{C}|\boldsymbol{\theta})}\frac{p(\boldsymbol{\theta})}{p(\mathcal{D})}d\boldsymbol{\theta} \\
&\qquad\qquad \text{(by Eq. (20) and rearranging terms)} \\
&= \log \int \frac{p(\boldsymbol{\theta}|\mathcal{D},\mathcal{F})}{p(\mathcal{C}|\boldsymbol{\theta})}d\boldsymbol{\theta} - \log \int \frac{p(\boldsymbol{\theta}|\mathcal{D})}{p(\mathcal{C}|\boldsymbol{\theta})}d\boldsymbol{\theta}. \qquad \text{(Bayes' rule)}
\end{aligned}
\tag{21}
$$

By doing so, instead of computing $p(\boldsymbol{\theta}|\mathcal{D}\setminus\mathcal{C})$ for every possible subset $\mathcal{C}$, the "predictive approach" only requires computing $p(\boldsymbol{\theta}|\mathcal{D},\mathcal{F})$ once. As shown in the main text, with (approximations of) the two posteriors $p(\boldsymbol{\theta}|\mathcal{D})$ and $p(\boldsymbol{\theta}|\mathcal{D},\mathcal{F})$ at hand, the log density ratio $r(\mathcal{C})$ can be efficiently approximated by Monte Carlo and/or further approximations described in the main text that employ Taylor expansions. This approach is "predictive" in the sense that the influence of candidate set $\mathcal{C}$ is evaluated by computing "predictions" $p(\mathcal{C}|\boldsymbol{\theta})^{-1}$ on them using the two posterior distributions, which is different from existing approaches that compute predictions on $\mathcal{F}$ using approximations to the modified posterior $p(\boldsymbol{\theta}|\mathcal{D}\setminus\mathcal{C})$.

### A.3 Objective for EWC-influence

Recall in the main text the first-order Taylor series approximation to $r(\mathcal{C})$ is

$$r(\mathcal{C}) \approx \sum_{\boldsymbol{z}\in\mathcal{C}} \hat{r}(\boldsymbol{z}), \quad \hat{r}(\boldsymbol{z}) = \mathbb{E}_{p(\boldsymbol{\theta}|\mathcal{D})}[\log p(\boldsymbol{z}|\boldsymbol{\theta})] - \mathbb{E}_{p(\boldsymbol{\theta}|\mathcal{D},\mathcal{F})}[\log p(\boldsymbol{z}|\boldsymbol{\theta})].$$

Therefore the "predictive approach" for computing $r(\mathcal{C})$ as well as the approximated form require the computation of (approximate) posteriors $p(\boldsymbol{\theta}|\mathcal{D})$ and $p(\boldsymbol{\theta}|\mathcal{D},\mathcal{F})$. This can be achieved using continual learning: we assume the model has been trained on $\mathcal{D}$ and an approximation $q(\boldsymbol{\theta})\approx p(\boldsymbol{\theta}|\mathcal{D})$ has been obtained. Then the current task for continual learning is to adapt the trained model on the failure cases $\mathcal{F}$, which leads to an adapted approximation $q_{+\mathcal{F}}(\boldsymbol{\theta})\approx p(\boldsymbol{\theta}|\mathcal{D},\mathcal{F})$.

Elastic Weight Consolidation (EWC) [10] is a continual learning algorithm that can be interpreted as updating the maximum a posteriori (MAP) approximation to the posterior given new tasks. To see this, first

in this approach the $q$ posteriors are assumed to be delta measures. In other words, $q(\boldsymbol{\theta}) = \delta(\boldsymbol{\theta} - \hat{\boldsymbol{\theta}})$ where $\hat{\boldsymbol{\theta}}$ is the parameters of the trained model, and $q_{+\mathcal{F}}(\boldsymbol{\theta}) = \delta(\boldsymbol{\theta} - \hat{\boldsymbol{\theta}}_{+\mathcal{F}})$ where as we shall see $\hat{\boldsymbol{\theta}}_{+\mathcal{F}}$ is the parameter obtained by running EWC using $\mathcal{F}$. With these assumptions, the EWC-influence is defined as:

$$r(\mathcal{C}) \approx \sum_{\boldsymbol{z} \in \mathcal{C}} \hat{r}(\boldsymbol{z}) \approx \sum_{\boldsymbol{z} \in \mathcal{C}} \tilde{r}(\boldsymbol{z}), \quad \tilde{r}(\boldsymbol{z}) = \log p(\boldsymbol{z}|\hat{\boldsymbol{\theta}}) - \log p(\boldsymbol{z}|\hat{\boldsymbol{\theta}}_{+\mathcal{F}}).$$

It remains to discuss the optimisation procedure for obtaining $\hat{\boldsymbol{\theta}}_{+\mathcal{F}}$. As motivated, we consider MAP approximations to the posterior, which seeks to find the maximum of the log posterior $\log p(\boldsymbol{\theta}|\mathcal{D},\mathcal{F})$. Notice that by Bayes' rule:

$$p(\boldsymbol{\theta}|\mathcal{D},\mathcal{F}) \propto p(\mathcal{F}|\boldsymbol{\theta})p(\boldsymbol{\theta}|\mathcal{D}) \quad \Rightarrow \quad \log p(\boldsymbol{\theta}|\mathcal{D},\mathcal{F}) = \log p(\mathcal{F}|\boldsymbol{\theta}) + \log p(\boldsymbol{\theta}|\mathcal{D}) + \text{constant}.$$

Computing the first term $\log p(\mathcal{F}|\boldsymbol{\theta})$ in the MAP objective is straightforward given the i.i.d. modelling assumption. For the second term, as $\log p(\boldsymbol{\theta}|\mathcal{D})$ is intractable, the EWC approach constructs a Laplace approximation to it by assuming the trained model parameter $\hat{\boldsymbol{\theta}}$ as a MAP point estimate of $p(\boldsymbol{\theta}|\mathcal{D})$. In detail, a Laplace approximation to the posterior is

$$\log p(\boldsymbol{\theta}|\mathcal{D}) \approx \frac{1}{2}(\boldsymbol{\theta} - \hat{\boldsymbol{\theta}})^{\top} \boldsymbol{H}[\log p(\hat{\boldsymbol{\theta}}|\mathcal{D})](\boldsymbol{\theta} - \hat{\boldsymbol{\theta}}) + \text{constant}. \tag{22}$$

where $\boldsymbol{H}[f(\boldsymbol{\theta})]$ denotes the Hessian matrix of a twice-differentiable function $f(\boldsymbol{\theta})$ with respect to parameters $\boldsymbol{\theta}$. Further decomposing the Hessian term yields:

$$\boldsymbol{H}[\log p(\hat{\boldsymbol{\theta}}|\mathcal{D})] = \boldsymbol{H}[\log p(\mathcal{D}|\hat{\boldsymbol{\theta}}) + \log p(\hat{\boldsymbol{\theta}}) - \log p(\mathcal{D})] \tag{23}$$

$$= \underbrace{\boldsymbol{H}[\log p(\mathcal{D}|\hat{\boldsymbol{\theta}})]}_{\approx -N\hat{\boldsymbol{F}}_{\hat{\boldsymbol{\theta}}}} + \underbrace{\boldsymbol{H}[\log p(\hat{\boldsymbol{\theta}})]}_{\text{Prior term}} \tag{24}$$

where $N$ is the total number of samples in $\mathcal{D}$ and $\hat{\boldsymbol{F}}_{\hat{\boldsymbol{\theta}}}$ is an empirical estimate of the *Fisher information matrix*. Assuming the zero-mean isotropic Gaussian prior $p(\boldsymbol{\theta})$ with precision $\lambda\boldsymbol{I}$, the second term becomes $\boldsymbol{H}[\log p(\hat{\boldsymbol{\theta}})] = -\lambda\boldsymbol{I}$. Substituting these results back into Eq. (22) gives us:

$$\log p(\boldsymbol{\theta}|\mathcal{D}) \approx -\frac{N}{2}(\boldsymbol{\theta} - \hat{\boldsymbol{\theta}})^{\top} \hat{\boldsymbol{F}}_{\hat{\boldsymbol{\theta}}}(\boldsymbol{\theta} - \hat{\boldsymbol{\theta}}) - \frac{\lambda}{2}||\boldsymbol{\theta} - \hat{\boldsymbol{\theta}}||_2^2 + \text{constant}. \tag{25}$$

Combining terms, the optimisation task for EWC adaptation using failure set is:

$$\hat{\boldsymbol{\theta}}_{+\mathcal{F}} = \underset{\boldsymbol{\theta}}{\arg\max} \log p(\mathcal{F}|\boldsymbol{\theta}) - \frac{N}{2}(\boldsymbol{\theta} - \hat{\boldsymbol{\theta}})^{\top} \hat{\boldsymbol{F}}_{\hat{\boldsymbol{\theta}}}(\boldsymbol{\theta} - \hat{\boldsymbol{\theta}}) - \frac{\lambda}{2}||\boldsymbol{\theta} - \hat{\boldsymbol{\theta}}||_2^2, \tag{26}$$

which is as presented in the main text. We further note that in the original EWC formulation, the empirical Fisher information is further simplified to contain *diagonal* entries only — with millions of model parameters as is typically the case with neural networks, as computing the full matrix is expensive.

### A.4 Objective for EWC-deletion

Assuming that the detrimental datapoints $\mathcal{C} \subset \mathcal{D}$ have been identified, the next step for model repairment is *treatment* where we seek to remove the influence of $\mathcal{C}$ to the model. This is done by computing (an approximation to) $p(\boldsymbol{\theta}|\mathcal{D}\backslash\mathcal{C})$ which can also be done via continue learning. To see this, we can show by using Bayes' rule and Eq. (20) again,

$$\log p(\boldsymbol{\theta}|\mathcal{D}\backslash\mathcal{C}) = \log \frac{p(\mathcal{D}\backslash\mathcal{C}|\boldsymbol{\theta})p(\boldsymbol{\theta})}{p(\mathcal{D}\backslash\mathcal{C})} = \log \frac{1}{p(\mathcal{C}|\boldsymbol{\theta})} + \log \underbrace{\frac{p(\mathcal{D}|\boldsymbol{\theta})p(\boldsymbol{\theta})}{p(\mathcal{D})}}_{=p(\boldsymbol{\theta}|\mathcal{D})} + \log \frac{p(\mathcal{D})}{p(\mathcal{D}\backslash\mathcal{C})}. \tag{27}$$

This means that finding the MAP estimate of $p(\boldsymbol{\theta} \mid \mathcal{D} \backslash \mathcal{C})$ is equivalent to maximising $-\log p(\mathcal{C} \mid \boldsymbol{\theta}) + \log p(\boldsymbol{\theta} \mid \mathcal{D})$ w.r.t. $\boldsymbol{\theta}$. With further approximation to the posterior $p(\boldsymbol{\theta} \mid \mathcal{D})$ using the Laplace method as discussed in section A.3, one can write the optimisation task for EWC-deletion as presented in the main text, namely:

$$\hat{\boldsymbol{\theta}}_{-\mathcal{C}} = \underset{\boldsymbol{\theta}}{\arg\max} -\log p(\mathcal{C}|\boldsymbol{\theta}) - \frac{N}{2}(\boldsymbol{\theta} - \hat{\boldsymbol{\theta}})^{\top} \hat{\boldsymbol{F}}_{\hat{\boldsymbol{\theta}}}(\boldsymbol{\theta} - \hat{\boldsymbol{\theta}}) - \frac{\lambda}{2}||\boldsymbol{\theta} - \hat{\boldsymbol{\theta}}||_2^2. \tag{28}$$

# B  Additional Results

## B.1   Sample Efficiency of Cause Identification

Fig. 6 compares the sample efficiency of different approaches for cause identification on MNIST and CIFAR-10 datasets with annotation noise. More specifically, we want to understand how many failure cases in the query set $\mathcal{F}_q$ we need to see for cause identification to work, i.e., how performance of each approach varies as we reduce the size of the query set. Here, all detrimental examples i.e., $\tilde{r}(\boldsymbol{z}) < 0$ are removed from the training data and the corresponding metrics are measured. EWC-influence approach performs the best in terms of precision, recall, and accuracy on holdout failure set $\mathcal{F}_h$ even as we decrease the size of the query set. All approaches exhibit a stable behavior w.r.t the size of query set. The biggest drop occurs in accuracy on the remaining test set $\mathcal{T} \setminus \mathcal{F}$: EWC-influence presents a stronger degradation in small sample regimes (around 5% of the original failure set) but without dropping below other approaches.

For the failure accuracy on MNIST, even when the query set is as small as 5% of the original one ($\approx 20$ examples), all approaches display improvement over the semi-oracle that is trained after removing all instances of annotation errors: this means that all methods, in a sample-efficient manner, not only correct the synthetically added annotation errors, but they also remove other harmful examples that are naturally present to begin with.

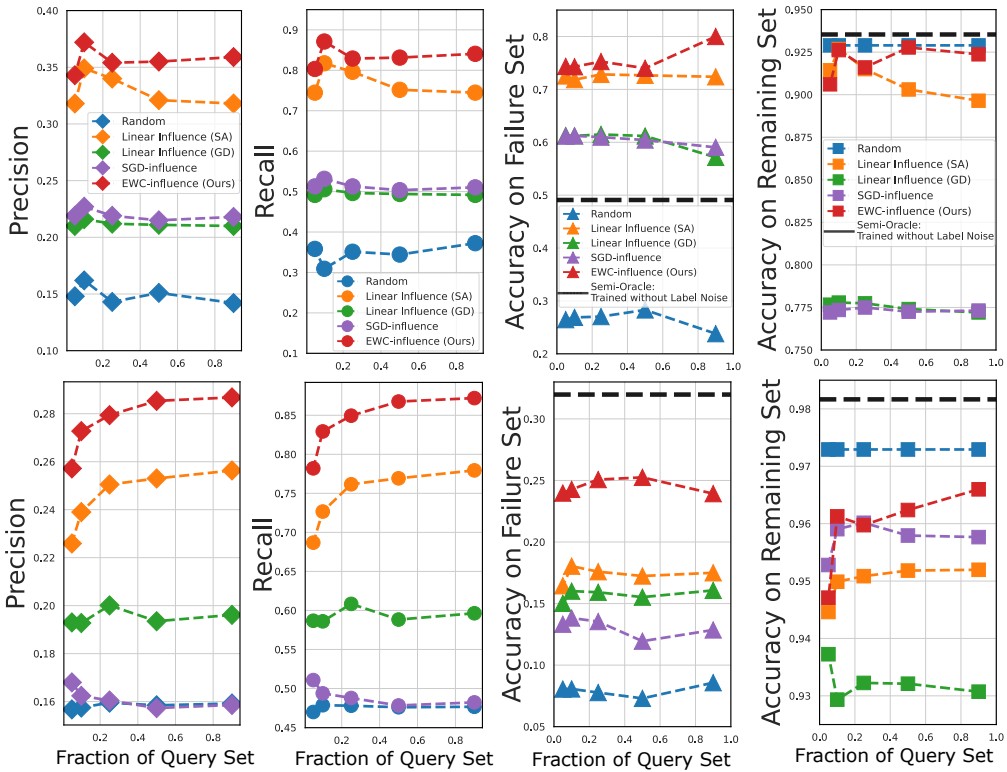

Figure 6: Comparison of sample efficiency for cause identification performance on (top) MNIST and (bottom) CIFAR-10 datasets with class-dependent annotation noise. On the x-axis of each sub-figure, we vary the size of the query failure set $\mathcal{F}_q$ used for cause identification. From left to right, (a) precision, (b) recall, (c) accuracy on holdout failure set $\mathcal{F}_h$, and (d) accuracy on the remaining test set.

## B.2 Speed Comparison

Table 1 shows the total computation time of the proposed EWC-influence and other baselines for cause identification on a single Tesla K80 GPU with 12GB of RAM. Note that we rely on the publicly available implementation of SGD-influence[3], and that we have implemented our own version of linear influence functions in Pytorch. Overall, EWC-Influence is either as fast or faster as other baselines, achieving one order of magnitude speed boost compared to SGD-influence in all cases. For CIFAR10 where a considerably larger base model is used, EWC-influence is consistently faster than other approaches, around twice faster than linear influence methods. For MNIST, EWC-influence attains similar computation time as linear influence approaches.

Table 1: Comparison of total computation time for cause identification.

| Experiment | Method | Time (s) ( MNIST ) | Time (s) (CIFAR10) |
|---|---|---|---|
| Label Noise | Linear Influence (SA) | 16.4 | 1322.4 |
| | Linear Influence (GD) | 11.2 | 996.2 |
| | SGD-Influence | 185.1 | 8301.1 |
| | EWC-Influence (Ours) | **9.8** | **496.3** |
| Random Input Noise | Linear Influence (SA) | 10.8 | 1141.1 |
| | Linear Influence (GD) | 15.7 | 978.4 |
| | SGD-Influence | 188.2 | 8285.6 |
| | EWC-Influence (**Ours**) | 11.0 | **527.5** |
| Adversarial Poisoning | Linear Influence (SA) | 14.3 | 1312.0 |
| | Linear Influence (GD) | **10.0** | 984.2 |
| | SGD-Influence | 177.29 | 7803.7 |
| | EWC-Influence (**Ours**) | 14.5 | **528.9** |

## B.3 Qualitative Results

**Annotation noise.** In the main text, Fig. 2(d) shows examples with annotation noise ranked as most harmful according to EWC-influence. Similarly, Fig. 7 shows examples ranked as least harmful according to EWC-influence. All of these examples correspond to non-corrupted examples with correct labels.

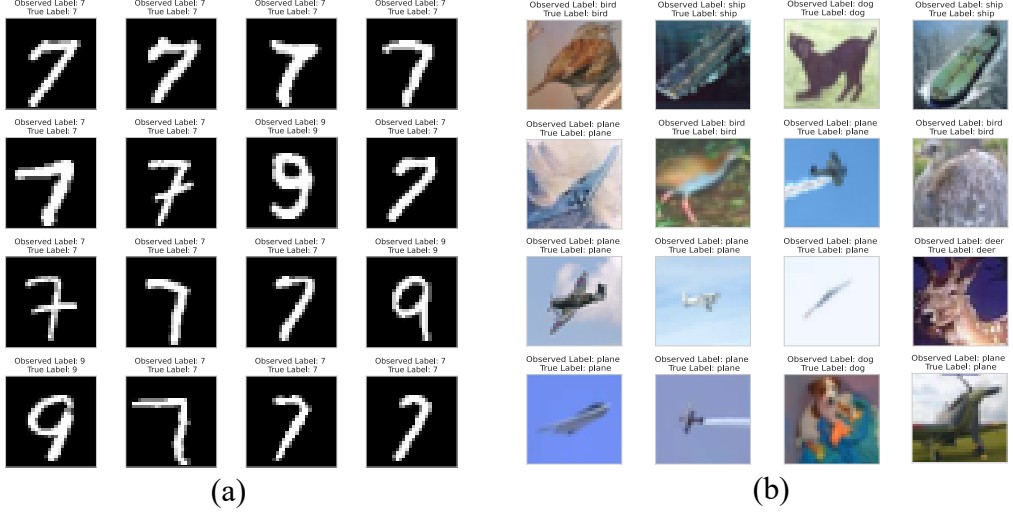

Figure 7: Examples of 16 **least** harmful examples for MNIST and CIFAR-10 with annotation noise as ranked by EWC-influence. None of the selected examples were corrupted with annotation noise.

---

[3]https://github.com/sato9hara/sgd-influence

**Random input noise.** Fig. 8 shows examples that are ranked highest (most harmful) by EWC-influence for datasets contaminated with random input noise. Recall that the target classes of the input noise are 1, 6, 7, 9 for MNIST and plane, bird, cat, dog for CIFAR10, and the rest of the images are free of such noise. First of all, we observe that the most detrimental examples belong to the non-target classes. As shown in the main text, while the input noise itself may not harm the performance of the model by much, the sample size of clean images in the target classes is still smaller as a result of noise injection—such group imbalance can be rectified by sub-sampling the dominant group, for example, by removing those identified detrimental data points. Secondly, many of them appear to be ambiguous instances in non-target classes. It is worth noting that for MNIST, one of the identified examples is interestingly a real instance of 3 that is incorrectly labelled as 5, which is also reported by Northcutt *et al.*[47].

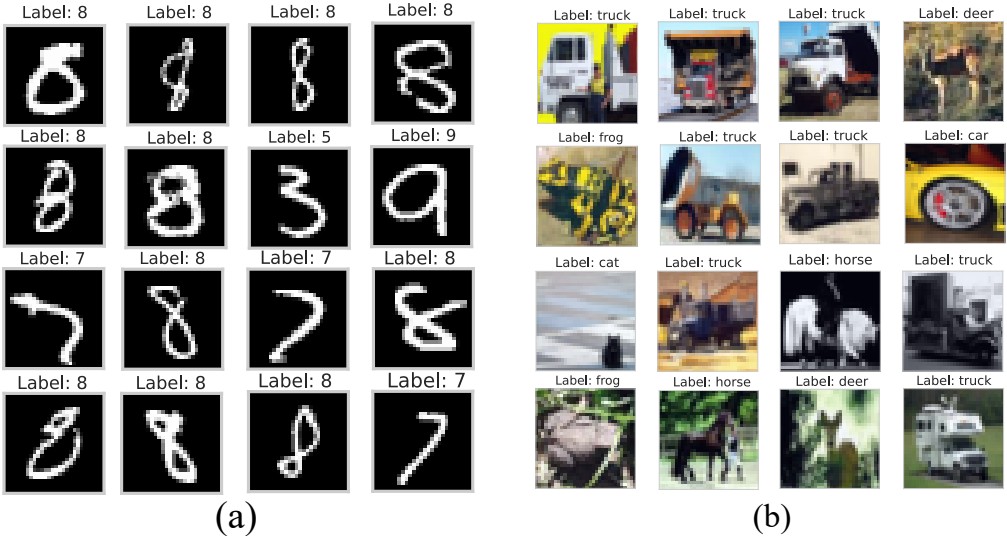

Figure 8: Examples of 16 most harmful examples for MNIST and CIFAR-10 with random input noise as ranked by EWC-influence. None of the selected examples were corrupted with random input noise.

### B.4   Treating Models by Forgetting Detrimental Past and Learning from Present Mistakes

While the primary goal of this work is to investigate how many prediction errors can be remedied by identifying harmful training data and removing them, one could alternatively use the labelled failure cases directly to adapt the model. Fig. 9 shows our preliminary results where we compare the deletion based methods to an approach that fine-tunes the model directly on the failure query set $\mathcal{F}_q$ with an L2-norm based locality constraint $\|\theta - \hat{\theta}\|_2^2$ [18] and its combination with the best deletion-based approach, that is, fine-tuning on the corrected dataset $\mathcal{D}\backslash\mathcal{C}$. As with other experiments, early stopping is performed based on the loss on a portion of the query set $\mathcal{F}_q$. We see that while fine-tuning on $\mathcal{F}_q$ (black points) leads to a higher accuracy (on the holdout failure set $\mathcal{F}_h$) than fine-tuning on $\mathcal{D}\backslash\mathcal{C}$ (yellow points), the accuracy on the remaining test set $\mathcal{T}\backslash\mathcal{F}$ is worse, even with the weight constraint — by 6% on MNIST and 21% on CIFAR-10. This issue of over-fitting to the failure cases is also reported in recent works such as [20, 19]. Importantly, by combining the two approaches (brown points), we can attain the best trade-off between the failure and the maintenance accuracy, indicating the complementarity between the proposed data correction methods and such fine-tuning approaches.

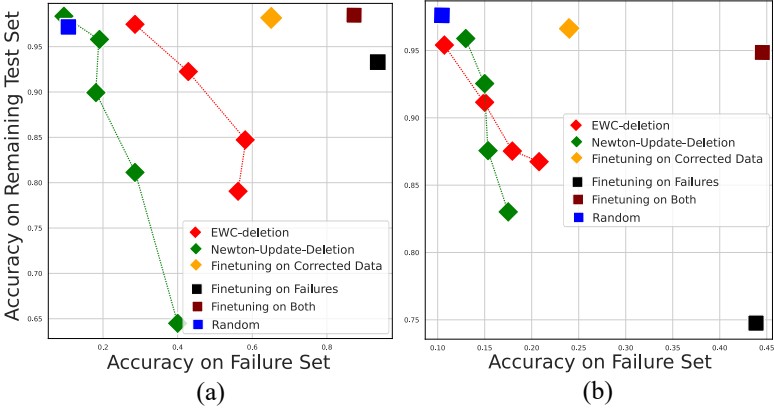

Figure 9: Comparison of deletion-based treatment methods and the direct fine-tuning on the failures on (a) MNIST and (b) CIFAR-10 datasets with noisy annotations.

## C    Experimental Details

**Datasets**    we perform our experiments on the MNIST digit classification task and the CIFAR-10 object recognition task. The MNIST dataset consists of $60{,}000$ training and $10{,}000$ testing examples, all of which are $28 \times 28$ grayscale images of digits from $0$ to $9$ (10 classes). The CIFAR-10 dataset consists of $50{,}000$ training and $10{,}000$ testing examples, all of which are $32 \times 32$ coloured natural images drawn from $10$ classes. Both datasets are preprocessed by subtracting the mean, but no data augmentation is used. For MNIST, to make the task more challenging, we randomly select 3000 examples from the training split and train the base models while the entire test set is used for evaluation.

**Architecture Details**    For MNIST, the base classifier was defined as a CNN architecture comprised of $4$ convolution layers, each with $3 \times 3$ kernels follower by Relu. The number of kernels in respective layers are $\{32,32,64,64\}$. After the first two convolution layers, we perform $2 \times 2$ max-pooling, and after the last one, we further down-sample the features with Global Average Pooling (GAP) prior to the final fully connected layer. For CIFAR-10, we used a 50-layer ResNet [49].

**Optimisation**    For all experiments, we employ the same training scheme unless otherwise stated. We optimize parameters using Adam [46] with initial learning rate of $10^{-3}$ and $\beta = [0.9, 0.999]$, with minibatches of size $64$ and train for max $100$ epochs with early stopping with a patience of 5, that is, training is stopped after 5 epochs of no progress on the validation set (10% of the training set). For computing both EWC-influence and EWC-deletion, we also employed the same training scheme but applied early stopping based on the performance on a validation split (10%) of the failure query set $\mathcal{F}_q$.

In Fig. 5 in Sec. 4.2, we present the performance of Newton update removal and EWC-deletion (ours) with different hyper-parameter settings. For Newton-update deletion [9], we vary the step size $\gamma > 0$ of the gradient ascent by scaling the second term in Eq. (16). For EWC-deletion (our method), we vary the amount of weight regularisation — the second term in Eq. (17) — by scaling it by $2/\gamma N$ where $N$ denotes the number of training datapoints, and $\gamma > 0$. We also set the strength of the prior term to $\lambda = 0$. We run both methods for different values of $\gamma$ in the range $[0.01, 0.05]$.

## D   Enlarged Figures

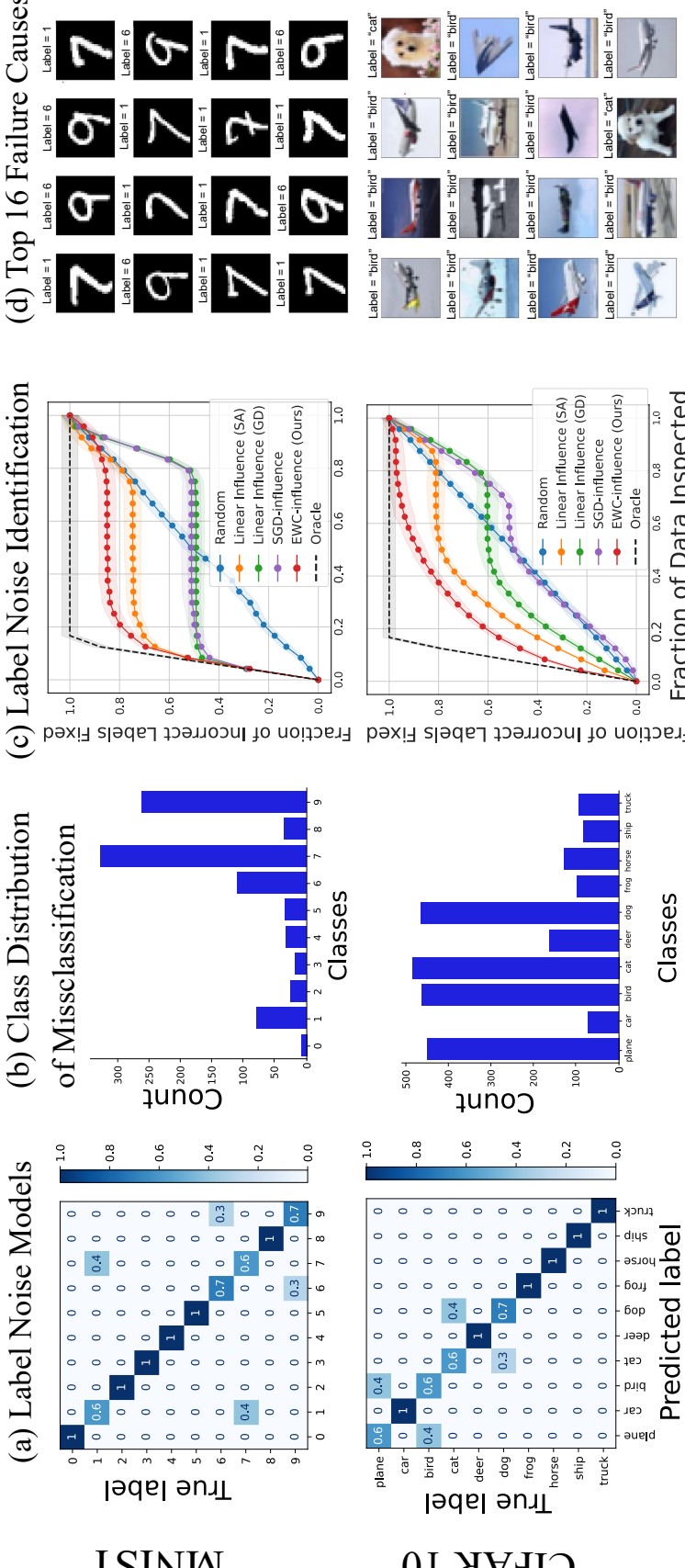

Figure 10: Results on cause identification in the presence of annotation noise. (a) shows the confusion matrices used to simulate class-dependent label noise on MNIST and CIFAR-10. (b) shows the class distribution of the misclassified examples for a single run. (c) plots how much of the identified causes match the samples with incorrect labels for different approaches. (d) shows the top 16 causes of the failures as ranked by EWC-influence. The shade represents the standard deviation computed from 5 different runs.

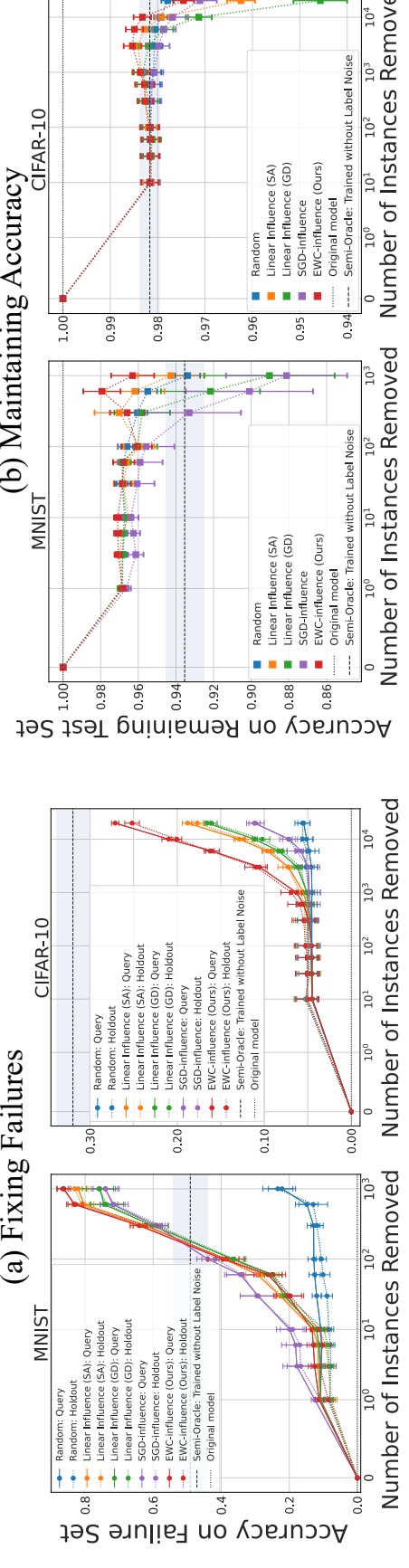

(a) Fixing Failures

(b) Maintaining Accuracy

Figure 11: Comparison of the quality of identified causes in the presence of annotation noise. The impact of gradually removing samples in MNIST and CIFAR-10 datasets in the order of influence values $r(z)$ are measured on the failure sets (holdout and query) in (a), and on the remaining test set in (b). We note that in (b), accuracy values start at 1.0 as they are calculated on the set of test samples on which the original model makes correct predictions. We also plot the performance of another reference ("semi-oracle") that is the original model fine-tuned on the training data without the label noise instances. The means and standard deviations of all quantities are calculated over 5 different runs.

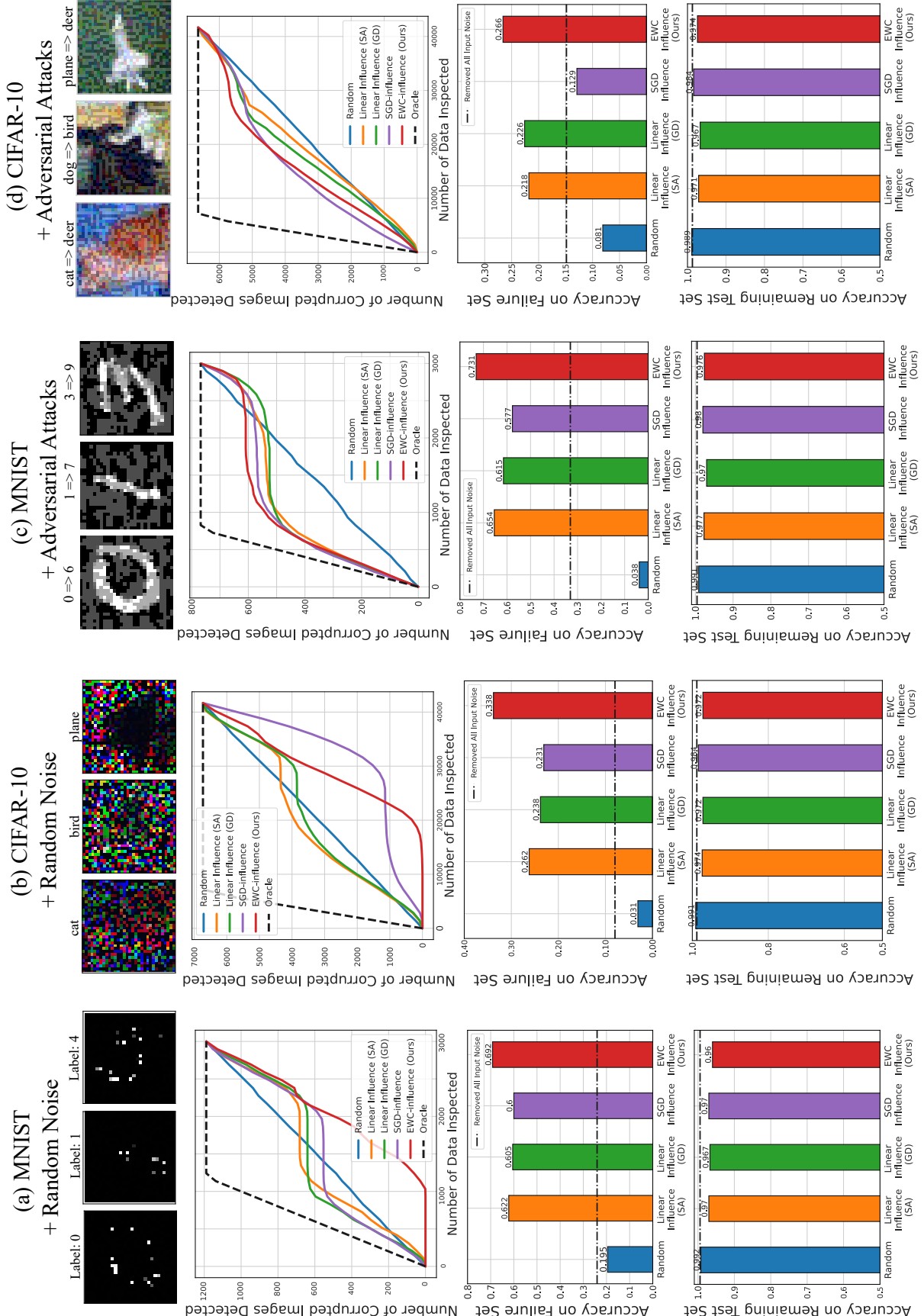

Figure 12: Results on cause identification in the presence of different input noises. From top to bottom, we show i) examples of corrupted samples (synthetic proxy for potential causes of failure), ii) how many of the identified causes correspond to samples corrupted with input noise, iii) and iv) performance in failure holdout set $\mathcal{F}_h$ and remaining test set when removing the top 1000/20000 identified causes in MNIST/CIFAR-10. The influence values are calculated with respect to 50% of test-time failure cases that belong to the classes that suffer from input noise. EWC-Influence identifies "harmful" (adversarial) input noise better than random while avoiding "harmless" (random) input noise.