# OpenReview forum: "Repairing Neural Networks by Leaving the Right Past Behind"
_NeurIPS.cc/2022/Conference — NeurIPS 2022 Accept_

### Official Review · Reviewer_7Beq · 2022-07-04

**Rating:** 6
**Confidence:** 3
**Soundness:** 3 good
**Presentation:** 2 fair
**Contribution:** 2 fair

**Summary:**

The authors propose a Bayesian continual learning approach which aims to remove memories from a neural networks weights if those memories stem from data which ultimately hurts the performance of the model on some defined failure set.

**Questions:**

- What are the concrete improvements over [2] which has a similar deletion approach?

- How does this approach compare to dataset condensation? It seems like a method of dataset condensation for a specific purpose.

- What is the underlying model which is used for CIFAR10? It looks like the 'original model' in figure 3 gets 100% accuracy, but the appendix quotes using ResNet50 from [1] where the a 50 layer network should give about 7% error rate.

- Were any of these results actually performed on a BNN with a distribution over the weights, or was the BNN interpretation only used for purposes of derivation?

- The legends in all charts in the main text are quite hard to read because of their small size, could the size of the legends and axes be increased?

- I find the notation and derivation of equations 5 and 6 to be too terse, and it took some effort to work out why they are true. Could a full derivation be included for the reader?

# References

[1] He, K., Zhang, X., Ren, S., & Sun, J. (2016). Deep residual learning for image recognition. In Proceedings of the IEEE conference on computer vision and pattern recognition (pp. 770-778).

[2] Guo, C., Goldstein, T., Hannun, A., & Van Der Maaten, L. (2019). Certified data removal from machine learning models. arXiv preprint arXiv:1911.03030.


**Limitations:**

Yes, the authors have offered some limitations in Section 2.

**Strengths And Weaknesses:**

# Strengths

- The method is well derived
- The experimental results show improvements over continual learning baselines.

# Weaknesses

- The datasets evaluated are quite small (MNIST, CIFAR-10). Given the method, it should be possible to use a pretrained model and apply it to a larger dataset such as ImageNet, correct? If that is the case, then couldn't a larger scale evaluation be done by taking the largest errors on the ImageNet validation set and applying the method and then measuring the difference between the Failure set and the remaining images in the ImageNet test set?

---

> ### Author Response · Authors · 2022-08-01
> **Response**
>
> Thank you for your review and suggestions on our work. We would like to address your questions/concerns below and are keen to follow up if you have any further questions.
>
>
> ----------
> > 1. The datasets evaluated are quite small (MNIST, CIFAR-10). Given the method, it should be possible to use a pretrained model and apply it to a larger dataset such as ImageNet, correct?
>
> Thank you for the suggestion. While we appreciate the value of evaluating the method on an additional large dataset, we strongly believe that its absence is not a show-stopper from the acceptance, and that our experiments demonstrate comprehensively the added value in a diverse set of different scenarios and hyper-parameters. We note our effort in the careful experimental design to investigate each component of the framework in detail. Controlled experiments with crafted label/input/adversarial noise are necessary for creating ``ground truths'' of failure causes and validating the quality of our methods.
>
> As mentioned in the summary post, we would also like to note that other reviewers appear to hold a similar view to us:
>  - **R4Kxq:** “... impressed by the experimental results of the method: particularly the results on identifying mislabeled examples and data poisoning …”
> - **Re7Kn:** “Empirical evaluation is pretty comprehensive”
>
> We would appreciate it if you could take this into account in the discussion phase, and we would like to engage in a conversation as much as you think is needed.
>
>
> ----------
> > 2. What are the concrete improvements over [2] which has a similar deletion approach?
>
> We believe the empirical improvements in deletion performance (shown in Fig.5) arise from the fact our approach (EWC-deletion) performs multiple steps of optimization while the Newton Update Deletion in [2] only performs a single step. We have clarified in a small paragraph (line 197 - 201) in blue in the revised manuscript to describe this in detail. Please let us know if this is still not clear.
>
>
> ----------
> > 3. How does this approach compare to dataset condensation? It seems like a method of dataset condensation for a specific purpose.
>
> Our understanding is that dataset condensation and our approach fall into different subsets of a more general **subset selection** problem. Dataset condensation aims to summarise data with a small set of *important* samples, independent of the models one may want to train on such data. Our approach differs from this in two ways: (1)**"different objective"**:  it aims to find a small set of *unimportant* samples in training data that contributed the most to some test-time failure cases; (2) **"model-specific"**: our subset selection procedure is tailored to the given model (e.g. different models may fail in different ways).
>
>
> ----------
> > 4. What is the underlying model which is used for CIFAR10? It looks like the 'original model' in figure 3 gets 100% accuracy, but the appendix quotes using ResNet50 from [1] where the a 50 layer network should give about 7% error rate.
>
> Sorry for the confusion. The accuracy starts at 100% in Fig. 3 by definition since these accuracy values are calculated on the set of test samples on which the original model makes correct predictions. So, when 0 examples are removed, it should be at 100% accuracy. We have now clarified this in the caption of Fig.3.
>
>
> ----------
> > 5. Were any of these results actually performed on a BNN with a distribution over the weights, or was the BNN interpretation only used for purposes of derivation?
>
> We used standard deterministic neural networks as the base architecture to ensure a fair comparison with other baseline methods such as Linear Influence, SGD-Influece, Newton Update Removal, etc. However, we note that our methods (EWC-influence, EWC-deletion) employ the Laplace approximation of the posterior distributions over the weights, which rely on the introduced Bayesian formalism.
>
>
> ----------
> > 6. The legends in all charts in the main text are quite hard to read because of their small size, could the size of the legends and axes be increased?
>
> Thank you for pointing this out, we have tried to increase the legends as best as we can in the revised version, but admittedly they still remain quite small. If the paper were to be accepted, we would have one extra page, which would allow us to accommodate this suggestion easily.
>
>
> ----------
> > 7. I find the notation and derivation of equations 5 and 6 to be too terse, and it took some effort to work out why they are true. Could a full derivation be included for the reader?
>
> Thank you for pointing this out. We have tried to improve the presentation in the revised manuscript, and I hope it is clear enough as a step-by-step derivation (lines 106-110).

---

> > ### Comment · Reviewer_7Beq · 2022-08-05
> > **Improvements over [2]**
> >
> > Thank you for the responses. I read through the paper again and looked at [2] and the main point which I was trying to ask is if there is any difference in the method you derived for cause identification (equations 2-7), and the method derived in [2].
> >
> > After looking at both papers again, it seems that [2] assumes the data for removal is known and your work in equations 2-7 allow for evaluating training set to find the subset for removal based on a specific failure set chosen by the user. Is that correct? Are there any other works related to deletion or unlearning which allow for the same or similar 'failure set' selection and 'problem set' identification?

---

> > > ### Author Response · Authors · 2022-08-05
> > > **Response To "Improvements over [2]"**
> > >
> > > Thank you for taking the time to take a further look into this connection.
> > >
> > > In short, **[2] and Eq. (2) - (7) are different as the former performs data deletion while the latter is concerned with cause identification**. Eq. (13) instead formalizes our approach to data removal, for which [2] is a specific example as summarised in lines 186 - 190. And yes, you are right that **[2] does not perform the cause identification**. However, there is a "dual" of [2] for cause identification, that is the *Linear Influence Function* proposed in [1], a *completely independent* piece of work (see Example 1 on pg 4). Our work reveals such a connection for the first time and abstracts it in such a way that any continual learning method can be adapted to both cause identification and data removal.
> > >
> > > Regarding your question on the prior works that perform both data deletion and failure set selection, **our work is the first to do so**. We provide a first general framework that connects data deletion and cause identification (under the Bayesian view of continual learning) --- we also highlight this connection in lines 177-181.
> > >
> > > Just to elaborate on the significance of this connection, **this unified view enables ones to take any continual learning method and extend it to both tasks**. An exemplary benefit of this connection, demonstrated in this work, is the adaptation of EWC to the model repairment setting, leading to novel techniques, and improvements in both tasks over the prior works (e.g. Newton update removal for deletion). We will describe this more clearly in the manuscript. Having one method for both brings coherence and simplicity to the framework, facilitating future research, and ultimately yielding improvements in performance.
> > >
> > > ---------
> > > References:
> > > [1] “Understanding Black-box Predictions via Influence Functions”, ICML 2017.

---

> > > > ### Comment · Reviewer_7Beq · 2022-08-07
> > > > **Thanks**
> > > >
> > > > Thanks for the added info. I have raised my score. I think that the point in this thread should be emphasized more in the final version because the significance of equations 2-7 can be lost in the rest of the work.

---

> > > > > ### Author Response · Authors · 2022-08-08
> > > > > **Thank you for the feedback**
> > > > >
> > > > > Thank you very much, and we will make sure to highlight the point of confusion you've raised in the final version. Please do let us know if you have any other questions that you would like us to address at all.

---

### Official Review · Reviewer_e7Kn · 2022-07-10

**Rating:** 6
**Confidence:** 2
**Soundness:** 3 good
**Presentation:** 2 fair
**Contribution:** 3 good

**Summary:**

This paper proposes a bayesian technique for determining the set of points that contribute to model errors and updating models to fix performance on the failure set while retaining performance. The key issue they overcome is determining the set of points that lead to errors is computationally hard, because you'd need to check every subset. Instead, they introduce a method that determines the set of points causing the errors in linear time, wrt dataset size. After they provide a technique to unlearn the datapoints in the failure set. Finally, they show their framework enables more efficient identification of the corrupt set in empirical evaluations over baselines such as random.



**Questions:**

- I'm a bit confused about the notation used with $p(F|D)$... This looks like the predictive distributon on the failure set given eq. 1, but its the predictive distribution of the model itself with the failure set. Could you clarify the usage here?
- I'm having a hard time following the Random Input Noise paragraph experiments. What is "salt and pepper"? It seems like ewc isn't detecting the corrupted images as efficiently as the baselines, what's going on here?
- Also, the bar plots for accuracy in fig. 4 have large ranges making it look like the accuracies are quite close, but for ewc for mnist for example, there's a 3 point different from random, which is very large for mnist. Is this method having sigificant effects on performance?
- Why does the test set accuracy for fig 3 start at 100%? It seems surprising these models getting 100% test set accuracy on these datasets?


**Limitations:**

The authors have done a sufficient job here.

**Strengths And Weaknesses:**

Strengths
- Interesting problem & quite relevant. The connections between influence functions are quite nice.
- Empirical evaluation is pretty comprehensive

Weaknesses:
- The notation usage is a bit dense and hard to follow in places. Often found myself having difficulty recalling what terms mean. Also, some terms seem like they might be a bit overloaded.
- The writing is hard to follow in places, and I found myself confused about what was going on several times while reading. This was particularly true in the empirical evaluations.

This paper is interesting overall and presents a seemingly useful method, but I'm having difficulty understanding the empirical evaluations in particular in several different places and found it quite hard to follow.

---

> ### Author Response · Authors · 2022-08-01
> **Response**
>
> Thank you for your encouraging review and your suggestions on clarity improvements. We will address your questions below.
>
>
> ----------
> > Q1. "I'm a bit confused about the notation used with $p(F|D)$... This looks like the predictive distributon on the failure set given eq. 1, but its the predictive distribution of the model itself with the failure set. Could you clarify the usage here?"
>
> Yes,  $p(F|D)$ is the (posterior) predictive distribution on the failure set, and in eq.(1), it signifies how well the failure set is captured by the current posterior distribution $p(\theta|D)$ and the likelihood function $p(F|\theta)$. This can be understood by rewriting the posterior predictive distribution as $p(F|D) = \int p(F|\theta) p(\theta|D) d\theta$. We have clarified this in the footnote on page 3.
>
>
> ----------
> > Q2. I'm having a hard time following the Random Input Noise paragraph experiments. What is "salt and pepper"? It seems like ewc isn't detecting the corrupted images as efficiently as the baselines, what's going on here?
>
> Thank you for pointing this out, we have added a description in line 297 of the revised manuscript. Salt-and-pepper noise is introduced by replacing randomly some pixels with extreme values, namely 0 (black pepper) or 255 (white salt).
>
>
> ----------
> > Q3. The bar plots for accuracy in fig. 4 have large ranges making it look like the accuracies are quite close, but for ewc for mnist for example, there's a 3 point difference from random, which is very large for MNIST. Is this method having significant effects on performance?
>
> Based on other experiments (Fig. 3), we believe the performance drop on the remaining test set from random to EWC is significant, although the differences amongst other influence function baselines are not. We stress, however, that such sacrifice is considerably smaller than the gain on the failure set (e.g., a 2-point difference from random).
>
>
> ----------
> > Q4. "Why does the test set accuracy for fig 3 start at 100%? It seems surprising these models getting 100% test set accuracy on these datasets?"
>
> The accuracy starts at 100% in Fig. 3 by definition since these accuracy values are calculated on the set of test samples on which the original model makes correct predictions (thus we refer to this as Accuracy on Remaining Test Set). So, when 0 examples are removed, it should be at 100% accuracy. We have clarified this in the caption of Fig. 3. Please let us know if this is still not clear.

---

> > ### Comment · Reviewer_e7Kn · 2022-08-05
> > **Thanks for the responses**
> >
> > Thanks for providing responses. These helped clarify my main points of confusion with the paper, so I'm upgrading my score.

---

> > > ### Author Response · Authors · 2022-08-08
> > > **Thank you for the response**
> > >
> > > Thank you for accounting for our response. We see, however, that your score still remains borderline.
> > > Please do let us know if you have any other remaining questions that we can address at all, and we would be eager to clarify and improve what we can.

---

### Official Review · Reviewer_4Kxq · 2022-07-11

**Rating:** 6
**Confidence:** 4
**Soundness:** 3 good
**Presentation:** 3 good
**Contribution:** 4 excellent

**Summary:**

The authors seek to find training points that most contribute to a model’s failures. They do so by using continual learning techniques to identify the training examples with the highest influence for model failures. They then again use continual learning to try to “forget” features learned by these training examples.

**Questions:**

I had some questions on the proof/theoretical motivation in Section 2.1:
- The i.i.d modeling assumption you make seems to be (for two disjoint portions $D, \mathcal{C}$ of the training dataset) that $P(\mathcal{D}, \mathcal{C} \mid \theta) = P(\mathcal{D} \mid \theta) P(\mathcal{C} \mid \theta)$. Can you walk through why this assumption is justified, and when it might not be? ($\mathcal{C}$ and $\mathcal{D}$ are not from the same distribution, since $\mathcal{C}$ are the examples that cause failures).
- In Example 1, you walk through how your method recovers linear influences. However, linear influences require taking a gradient step on the training example (so here z) and then measuring the change in prediction of the test example (so here $\mathcal{F}$). You seem to be doing the reverse. Are you recovering the transpose of the linear influence matrix?
- What is $z^*$ in line 140?


**Limitations:**

The authors did not address limitations of their work, or discuss negative societal impact. (If I missed it, please point it out in the author response)

**Strengths And Weaknesses:**

## Strengths
*Quality:* I was relatively impressed by the experimental results of the method: particularly the results on identifying mislabeled examples and data poisoning. This certainly seems like a cheaper method for isolating influential points than retraining.

*Originality/Significance:* The problem is relevant to the community, and the method seems to be sufficiently novel. Additionally, the framework for "forgetting" the features learned by particular datapoints is interesting.

## Weaknesses
*Clarity*: Section 2.1 was not clearly written (see questions below). It may be helpful to start with why, even though $\mathcal{F}$ is in the test set, this method requires taking gradient steps on $\mathcal{F}$.  Providing additional intuition on EWC in the main paper would be helpful as well (this information is currently in the Appendix, but could be moved up in a compressed form).

*Quality:* What happens when you apply this to an untouched CIFAR-10 dataset? Can you identify mislabeled examples within the dataset? Does your accuracy improve, or does performing continual learning in this setting reduce accuracy if there is no planted pathology?

---

> ### Author Response · Authors · 2022-08-01
> **Response**
>
> Thank you for your encouraging review and your suggestions on clarity improvements. We will address your questions in detail in the following.
>
> ----------
> > Q1.It may be helpful to start with why, even though $F$ is in the test set, this method requires taking gradient steps on $F$. Providing additional intuition on EWC in the main paper would be helpful as well (this information is currently in the Appendix, but could be moved up in a compressed form).
>
> Thank you for the suggestion. We have now provided an intuition on EWC in the main text (line 148 - 150 in blue).
>
>
> ----------
> >  Q2. What happens when you apply this to an untouched CIFAR-10 dataset? Can you identify mislabeled examples within the dataset?
>
> Thanks for asking this question. It is an interesting one. While we do not have results from this exact setting, we do have a close scenario with the random input noise in which the planted pathology does **not** affect the classification accuracy. As shown in Fig.8 in Appendix B,   in the absence of no extra pathology, we thus suspect that the identified causes would coincide with ambiguous instances or cases with real incorrect labels as you have also alluded to.
>
>
> ----------
> > Q3. The i.i.d modeling assumption you make seems to be (for two disjoint portions $D, C$ of the training dataset) that $p(D, C|\theta) = p(D|\theta)p(C|\theta)$ . Can you walk through why this assumption is justified, and when it might not be?
>
> The key assumption here is the statistical independence of the data samples. While this is a fairly standard assumption made for convenience in many lines of work, as you have pointed out, it may not hold in practice e.g., the acquisition of some samples may inform the acquisition of other samples in certain settings such as Bayesian optimization/active learning scenarios.
>
>
> ----------
> > Q4. In Example 1, you walk through how your method recovers linear influences. However, linear influences require taking a gradient step on the training example (so here z) and then measuring the change in the prediction of the test example (so here $F$ ). You seem to be doing the reverse. Are you recovering the transpose of the linear influence matrix?
>
> The sign difference arises from the difference that our work aims to measure the "negative" impact of data while the original form of Linear Influence function in [(Koh & Liang, ICML 2017)](https://arxiv.org/abs/1703.04730) was introduced to measure their "positive" impact. We have now clarified this in lines 139-141.
>
>
> ----------
> > Q5. What is $z^*$ in line 140?
>
> Apologies for the confusion. Here $z^*$ denotes a test sample of interest. In [(Koh & Liang, ICML 2017)](https://arxiv.org/abs/1703.04730), we realize that they denote this by $z_{test}$ --- we have now replaced this to be consistent with their work (line 139).

---

> > ### Comment · Reviewer_4Kxq · 2022-08-05
> > **Update**
> >
> > The authors have sufficiently addressed my concerns (I have raised my score accordingly). I agree with reviewer e7Kn here: the notation and explanation is still pretty dense, which makes it hard to follow. This can be fixed by
> > 1. Reducing notation as much as possible, and reminding the reader what different terms correspond to
> > 2. Split section 2.1 into several sub-steps (suggested split: lines 105 and 120). At the end of each sub-step, summarize what was accomplished in that substep. It's pretty easy to get lost otherwise.
> > 3. If space is a concern, move some of the examples to the appendix unless they were used in the experiments. Spend that space on explaining the core method more clearly.

---

> > > ### Author Response · Authors · 2022-08-08
> > > **Thank you for the suggestions**
> > >
> > > Thank you very much for the suggestions, and we will take these into account in the final version of the manuscript.
> > > Please let us know if you have any other questions.

---

### Author Response · Authors · 2022-08-01
**Summary**

We thank all reviewers for their constructive feedback. We summarize below the strength of the paper and the criticisms from the reviewers, and also bring to attention some issues that are noteworthy in our view:

------------
**Strengths & reviewers’ support**:
- Our framework for repairing models consists of two key components: 1) **"cause identification"**: a mechanism for identifying the ``causes'' in training data that are responsible for the given test-time failures and 2) **"treatment"**: adaptation method for fixing the model by removing information about them. Crucially, the two are connected under the Bayesian view of continual learning, which brings forth two **practical benefits**. Firstly, the framework **subsumes prior works** on influence function [[1]](https://arxiv.org/abs/1703.04730) and data deletion [[2]](http://proceedings.mlr.press/v119/guo20c.html) as specific examples, and our work **reveals their close connections and limitations**. Secondly, the generality of this novel viewpoint **allows translating any continual learning method into this new model repairment setting** and opens doors to future research. In particular, we show an exemplary application where importing Elastic Weight Consolidation [[3]](https://www.pnas.org/doi/10.1073/pnas.1611835114) to this new problem leads to **significant empirical enhancements** over the relevant baselines.

- All the reviewers acknowledge the relevance and novelty of our work:
> - **R4Kxq:** “The problem is relevant to the community, and the method seems to be sufficiently novel….”
>  - **Re7Kn:** “Interesting problem & quite relevant. The connections between influence functions are quite nice.”
>  - **R7Beq:** “The method is well derived, … show improvements over continual learning baselines”


------------
**Main issues raised by the reviewers**:
1. *Clarity*: Clarity is the main concern from **Reviewers 4Kxq, e7Kn**. In the revised version, we have tried our best to improve the presentation and highlighted the changes in blue.

2. *Insufficient Large-scale Experiments*: **Reviewer 7Beq** raises a lack of experiments on very large-scale datasets (e.g., ImageNet) as the main weakness. While the inclusion of such a dataset would be beneficial, we strongly believe that this should not be a show-stopper from the acceptance, and our work has convincingly demonstrated the values of our work through extensive experiments in a diverse set of different scenarios and hyper-parameters. **The other two reviewers (Reviewers 4Kxq, e7Kn) agree with us in this regard**:
 > - **R4Kxq:** “... impressed by the experimental results of the method: particularly the results on identifying mislabeled examples and data poisoning …”
> - **Re7Kn:** “Empirical evaluation is pretty comprehensive”

------------
**Mismatch between reviews & scores**:
Lastly, we would like to highlight that there seems to be a mismatch between the contents of the reviews and the provided scores. In particular, we notice that the main weaknesses raised by **R4Kxq, Re7Kn** are concerned with the clarity of the presentation, and their reviews appear to be overall much more positive than the provided borderline scores (4, 5 respectively). Now that we have tried our best to address their concerns, we would be grateful if the reviewers could revisit their evaluation. We are eager to discuss further and clarify as much as needed.

----------
**References:**
- [1] “Understanding Black-box Predictions via Influence Functions”, ICML 2017.
- [2] “Certified Data Removal from Machine Learning Models”, ICML 2020.
- [3] “Overcoming catastrophic forgetting in neural networks”, PNAS 2017

---

> ### Author Response · Authors · 2022-08-09
> **Post-Discussion Update**
>
> We would like to thank the reviewers for taking a look at our rebuttal/revised submission and engaging in discussions where appropriate. Below we summarise the two main outcomes of the discussion phase:
>
> 1. Technical concerns have been addressed in the rebuttal & the revised manuscript. All reviewers think that the overall framework is novel and important.
>
> 2. Remaining issues are the clarity of the method section (e.g., restructuring as per *Reviewer 4Kxq*'s suggestions) & the figure size, while otherwise, the reviewers think the other clarification questions have been addressed in the new manuscript. Given the camera-ready version allows 10 pages, if accepted, we can and will address the remaining presentation issues.

---

### Meta-Review · Area_Chair_7Per · 2022-08-20

**Recommendation:** Accept
**Confidence:** Certain

**Metareview:**

This paper studies the setting where some data points are contaminated and as a result, the learned method suffers from performance degradation. Authors extend an existing continual learning algorithm called Elastic Weight Consolidation and use it for identifying and removing data points that are harmful to performance. The experiments confirms that their proposed methods can indeed identify the harmful data points and repair the learned model. All reviewers find this paper interesting and they are in agreement about accepting the paper. However, they are also in agreement that the paper is hard to follow. I suggest authors to take reviewers suggestions into account and improve the presentation for the camera-ready version to make it easier for the community to benefit from and build on this paper.

**Award:**

No

---

### Decision · Program_Chairs · 2022-09-14

Accept